# Protein-Peptide Turnover Profiling reveals the order of PTM addition and removal during protein maturation

Henrik M. Hammarén ®[1,4] ✉, Eva-Maria Geissen ®[2,4], Clement M. Potel[1], Martin Beck ®[3] ✉ & Mikhail M. Savitski ®[1] ✉

Post-translational modifications (PTMs) regulate various aspects of protein function, including degradation. Mass spectrometric methods relying on pulsed metabolic labeling are popular to quantify turnover rates on a proteome-wide scale. Such data have traditionally been interpreted in the context of protein proteolytic stability. Here, we combine theoretical kinetic modeling with experimental pulsed stable isotope labeling of amino acids in cell culture (pSILAC) for the study of protein phosphorylation. We demonstrate that metabolic labeling combined with PTM-specific enrichment does not measure effects of PTMs on protein stability. Rather, it reveals the relative order of PTM addition and removal along a protein's lifetime—a fundamentally different metric. This is due to interconversion of the measured proteoform species. Using this framework, we identify temporal phosphorylation sites on cell cycle-specific factors and protein complex assembly intermediates. Our results thus allow tying PTMs to the age of the modified proteins.

Metabolic labeling coupled with mass spectrometry (MS) has become a mainstay of measuring protein turnover and degradation rates in cells. In cell culture experiments, arginine and lysine labeled with stable carbon and nitrogen isotopes are typically used in a method called pulsed (or dynamic) stable isotope labeling of amino acids in cell culture (pSILAC)[1,2]. Following labeling for a defined time, cells are lysed and proteins digested with trypsin, which cleaves after arginine and lysine residues, thus leaving each resulting peptide carrying at least a single labeled or unlabeled residue. MS is then used to measure the label incorporation rate for each identified peptide. In a steady-state system (or a steadily growing cell population, where the effect of cell growth can be accurately determined and subtracted from the data, see Supplementary Note 1), the incorporation rate equals the rate of clearance of the specific peptide from the system. In the case of a single protein species, which is cleared from the system by whole-protein degradation, the clearance rate equals the degradation rate. In this case, the degradation rate constant can be determined from the median clearance rate of all proteotypic peptides of the protein[1–4] (Fig. 1).

In reality, however, most eukaryotic proteins are present in more than a single species (or "proteoform"). While some of these arise from alternative splicing of transcripts, the vast majority of cellular proteoforms is thought to be defined by post-translational modifications (PTMs)[5], such as addition and removal of chemical groups (phosphorylation, acetylation, etc.) or proteolytic processing. These proteoforms often differ from one another only on a single amino acid residue (the one carrying the PTM), which can result in only a single proteoform-specific peptide after trypsin digestion for MS analysis. Importantly, the fate of different PTM-defined proteoforms is interlinked, as they are interconverted from one another, often in a reversible process of PTM writing and erasing. Crucially for metabolic labeling experiments, this interconversion can effectively "remove" a specific peptide from the system (by changing it to a differently modified peptide) by other means than degradation. As a

[1]European Molecular Biology Laboratory, Genome Biology Unit, 69117 Heidelberg, Germany. [2]European Molecular Biology Laboratory, Structural and Computational Biology Unit, 69117 Heidelberg, Germany. [3]Max Planck Institute of Biophysics, Department of Molecular Sociology, 60438 Frankfurt am Main, Germany. [4]These authors contributed equally: Henrik M. Hammarén, Eva-Maria Geissen. ✉e-mail: henrik.hammaren@embl.de; martin.beck@biophys.mpg.de; mikhail.savitski@embl.de

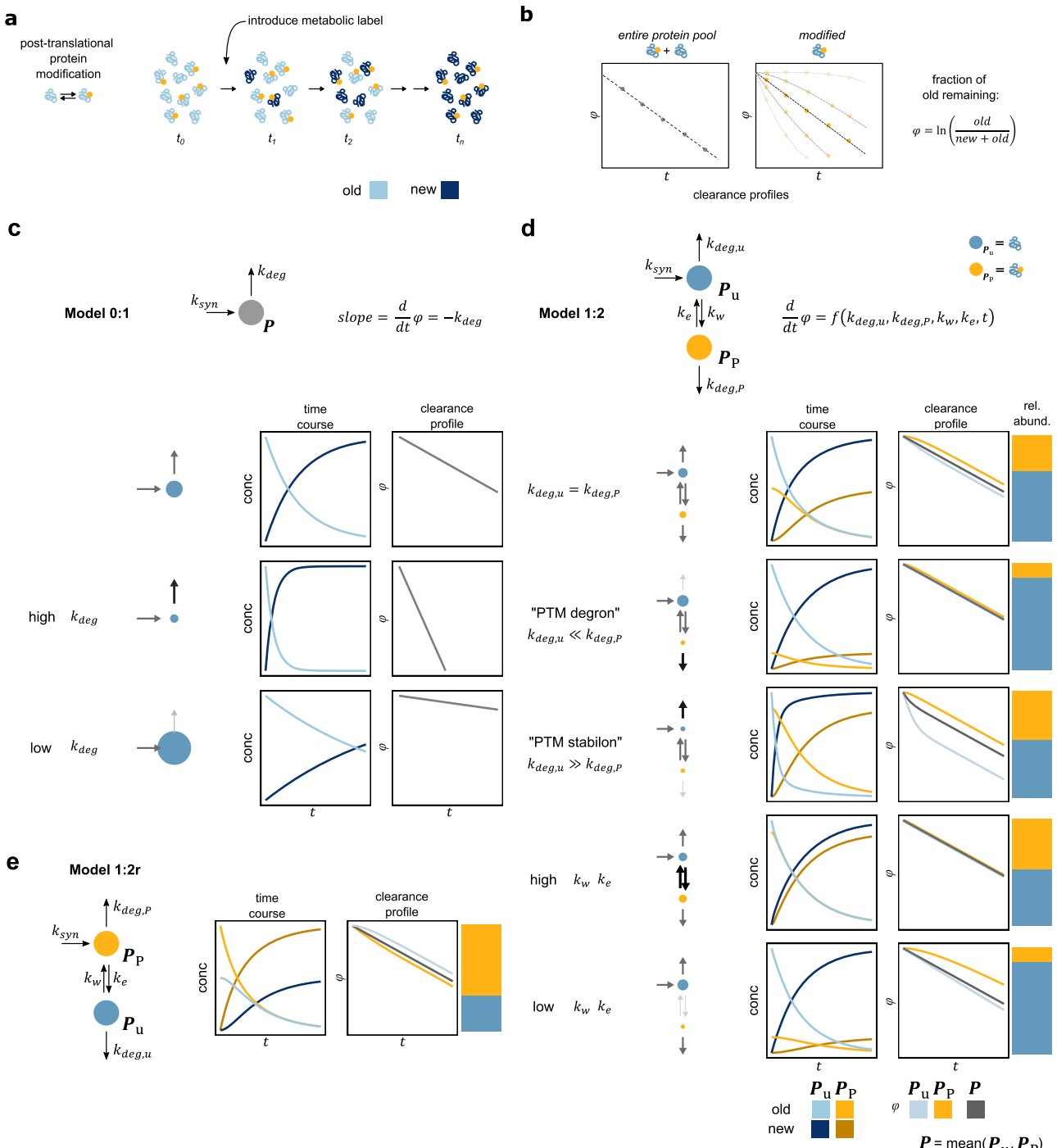

**Fig. 1 | Quantitative modeling of multi-species metabolic labeling experiments identifies limits and possibilities of the approach. a** In a steady-state system with interconverting species, such as proteins with or without a PTM, an introduced metabolic label equilibrates into both unmodified and modified protein species through writing and erasing of the PTM (yellow circle). **b** Previous pulse-labeling experiments have shown that clearance of modified peptides can differ substantially from the rest of the protein. Shown are archetypal clearance profiles (logarithm of fraction of old protein remaining, $\varphi$, over time, $t$) for the entire protein pool (gray) and the modified species (yellow). **c** In the simplest, single-species model for protein turnover, the slope of the clearance profile is directly defined by the protein degradation rate constant, $k_{\text{deg}}$. Models are named "Model x:y" (e.g., Model 0:1) indicating the number of modified species (x), and the number of species altogether (y). In the small cartoons, the size and opacity of the arrows reflect the magnitude of the rate constant. The size of the circle reflects the steady-state amount of the species. **d** Two-species model (Model 1:2) including protein modification. The slopes of the clearance profiles are complex functions of the model parameters (all but the synthesis rate $k_{\text{syn}}$), and change over time (see Supplementary Note 1 for full analytical description). In Model 1:2 clearance of the modified species $P_P$ can never be faster than the clearance of the entire protein pool $P$. Parameter combinations with distinct biological interpretation have practically indistinguishable clearance profiles. **e** Alternative (reverse, r) two-species model, Model 1:2r, in which a protein is synthesized as a modified species, allows faster clearance of the modified species $P_P$ compared to the entire protein pool $P$. The relative order of clearance profiles is defined by the order of species (or "wiring" of the modification network) during a protein's lifetime. See also https://apps.embl.de/pptop for an interactive web application of the models.

consequence, the relation between clearance rate and degradation rate becomes nontrivial.

Here, we examine this relation using theoretical considerations based on first principles and experimentally test our resulting hypotheses in the context of protein phosphorylation. We show, using quantitative kinetic modeling, that clearance rates measured by pSILAC-MS for different proteoforms are not straightforwardly defined by effects of the PTMs on a protein's proteolytic stability (i.e., the degradation rate). Rather, clearance rates and profiles are primarily defined by the relative order of addition (i.e., the network structure or "wiring") and affected by the kinetics of addition and removal of the modification itself. Thus, instead of readily giving information on protein stability per se, differences in clearance rates suggest hypotheses on the temporal ordering of modification events along the synthesis-maturation-degradation axis (i.e., age or lifetime) of each protein. We test these hypotheses by combining pSILAC with phosphoenrichment and peptide-level turnover analysis in a method we dub Protein-Peptide Turnover Profiling (PPToP, similar to DeltaSILAC[6], or Site-resolved Protein Turnover profiling, SPOT[7]). In accordance with our hypotheses, we find that the majority of phosphorylated peptides exhibit slower clearance than the respective protein median, characteristic of modifications occurring later in a protein's lifetime. Furthermore, we identify peptides with faster clearance, corresponding to known protein N-terminal maturation intermediate proteoforms. We corroborate our model by mutagenesis of 22 target proteins carrying 63 phosphosites. Using PPToP, we identify temporal proteoforms corresponding to phosphorylation/dephosphorylation events of cell cycle-specific factors as well as protein complex maturation intermediates. We further show that PPToP data with high time resolution can provide in cellulo kinetic rate parameters of PTM writing and erasing.

## Results

### Clearance profiles are primarily defined by the order of modification

In the case of a single pool of protein $P$ in a steady-state system, the rate of clearance of the old protein (equal but opposite in sign to the rate of incorporation of new label) measured in a pSILAC-MS experiment (Fig. 1a) is defined by a single parameter: the degradation rate constant of the protein, $k_{deg}$ (Fig. 1b, c). To understand how adding a separate interconvertible protein pool (such as a proteoform defined by a PTM) affects measured clearance rates, we built quantitative kinetic models of synthesis-modification-degradation networks (Fig. 1c–e). The simplest model consists of two species: the unmodified protein, $P_u$, and a modified protein, $P_P$. Assuming a protein is synthesized in an unmodified form and is then modified in a first-order reaction (see Supplementary Note 1 for full descriptions and underlying assumptions), we define a model that we term Model 1:2 (Fig. 1d), for consisting of a singly-modified species (1), and two species in total (2). In Model 1:2, the rates of clearance of $P_P$ and $P_u$ become a function of not only the degradation constants of both protein species ($k_{deg,u}$ and $k_{deg,P}$), but also of the rate constants of interconversion (rate constants of writing and erasing the PTM, $k_w$ and $k_e$, respectively). Comparing the clearance rates of $P_P$ and the entire protein pool $P$ ($=P_P + P_u$, measured in practice as the median clearance rate of all shared peptides), we find that $P_P$ exhibits slower clearance compared to $P$ independently of the parameters used (Supplementary Note 1, and Fig. 1d and Supplementary Fig. 1a). Importantly, this not only includes cases, where a PTM might affect a protein's proteolytic stability (so called "degrons", when $k_{deg,P} \gg k_{deg,u}$, or "stabilons", when $k_{deg,P} \ll k_{deg,u}$), but also when there is no change in a protein's proteolytic stability ($k_{deg,P} = k_{deg,u}$). Thus, a slower clearance rate for $P_P$ cannot be interpreted to, for example, imply a stabilizing effect for the PTM in question. This stems from the fact that in Model 1:2 new label is introduced first into $P_u$ (by synthesis) only from which it can subsequently enter into $P_P$ through

modification (Fig. 1d). Thus, the clearance rate of $P_P$ can at most equal the clearance rate of $P$ as seen for rapid interconversion of the species (Fig. 1d: high $k_w$ and $k_e$). Furthermore, for cases, where the entire protein pool $P$ has a linear clearance profile (which has been shown experimentally to be the case for most proteins[8], as well as constitutes the majority of theoretically achievable profiles, Supplementary Fig. 1b), the magnitude of the difference in clearance of $P$ and $P_P$ is primarily defined by the writing and erasing rate constants ($k_w$, $k_e$, see Supplementary Note 1 and Supplementary Fig. 1).

What does it take then, to produce faster clearance of $P_P$ compared to $P$, as has been observed experimentally before[6,7]? Arguably, the simplest solution is to reverse the relative order of the two species (Model 1:2r, Fig. 1e). This can biologically be interpreted to represent modification during or immediately after translation (i.e., cotranslationally)[9–12]. Analogously to Model 1:2, this causes clearance of $P_P$ to always be faster or equal to $P$. Thus, relative clearance (or the "clearance profile") is primarily defined by the relative order of appearance of the measured species with respect to protein synthesis, which is defined by the model structure, or "wiring".

Naturally, networks of protein modification could be expanded to be arbitrarily complex to represent different biological situations. With increasing complexity of the model (and a resulting increase in parameters), the flexibility of the model increases concomitantly. Thus, for instance, a three-species model can be devised (Model 1:3, see Supplementary Note 1 and Supplementary Fig. 8), where differences in clearance can also appear to be in line with differences in the degradation rate constants ($k_{deg,u}$ and $k_{deg,P}$). However, without prior knowledge of the shape of the network, these behaviors cannot be unambiguously distinguished from effects caused by the relative order of modification along a protein's lifetime using measured clearance rates of $P_P$ and $P$ alone (see Supplementary Note 1).

### Experimental detection of proteoforms with differing clearance rates

The theoretical considerations above let us formulate three key predictions: (i) the default, (and functionally least informative) expected behavior for a PTM is slower clearance, (ii) interconvertible species with faster clearance likely represent early (i.e., close to synthesis) intermediates in protein maturation, and (iii) relative differences in clearance rates are not predictive of effects on protein proteolytic stability. To test these predictions for the case of protein phosphorylation, we devised PhosphoProtein-Peptide Turnover Profiling (Phospho-PPToP) combining two-label pSILAC labeling with phosphoenrichment of peptides. We focused on early time points to enable observation of rapidly cleared transient species and sampled HeLa cells in 9 time points starting from 30 min after medium exchange (Fig. 2a). Following harsh lysis and trypsin digestion, phosphopeptides were enriched, and both the phospho-enriched eluate and the total input were prefractionated and measured using data-dependent acquisition (DDA) MS. We achieved high phosphoenrichment efficiency of 97%, and after filtering for reproducibility (Supplementary Fig. 2a) and presence in at least 2 replicates and 2 time points, we could quantify 67,393 unique modified peptides (covering 6749 gene names) carrying 10,765 unique phosphosites (2880 gene names), including 2644 unique phosphosites (1065 gene names) for which peptides were quantified both in an unphosphorylated and phosphorylated form. Quantifying clearance of pre-existing ("old") amino acids after medium exchange, we found that most unmodified peptides exhibit a relatively tight clearance distribution (Supplementary Fig. 2b). This is expected, since most proteins should turn over as entire polypeptides, and only peptides typical to non-majority proteoforms should show deviating clearance. Correspondingly, phosphopeptides exhibit more varied clearance compared to unmodified peptides with both faster and slower behaviors (Supplementary Fig. 2b), suggesting that the measured phosphorylated peptides indeed represent distinct

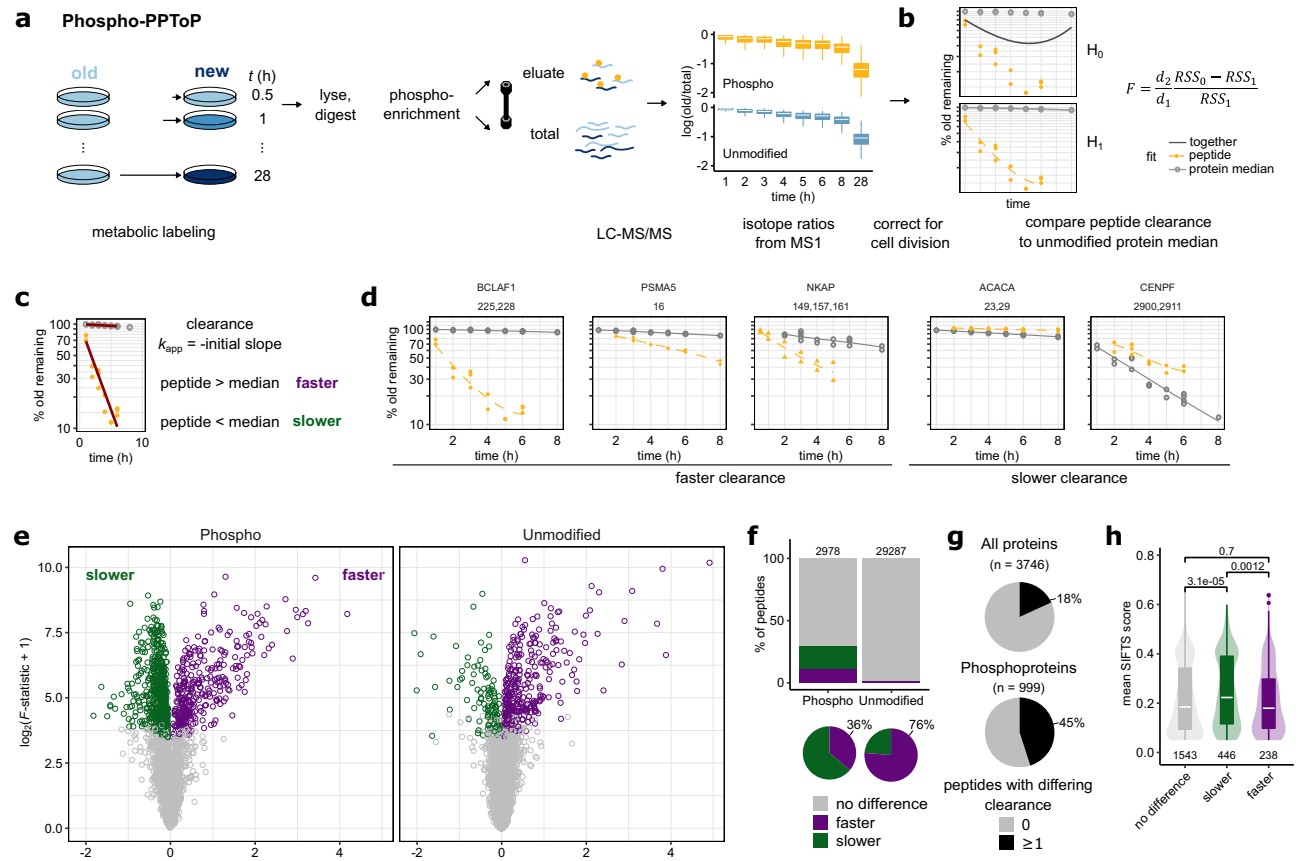

**Fig. 2 | Measuring peptide clearance with PPToP. a** Experimental setup. HeLa cells were grown in isotopically light (L)-labeled medium and pulsed for a specified time with heavy (H)-labeled medium (Replicates 1–3; L and H reversed for Replicate 4). After lysis and trypsin-digest, phosphorylated peptides were enriched using Fe-IMAC. Both the eluate enriched for phosphorylated peptides (phospho, in yellow) as well as the flow-through (unmodified, in blue) were measured with mass spectrometry. Boxplots show replacement of old with new labeled protein over time, y axis: median values over replicates on the protein level. Outliers have been omitted for clarity. **b** After correction for cell growth (see "Methods"), clearance traces for each peptide are compared to the trace of the median of all unmodified peptides (representing the entire protein pool *P*) to find peptides with significantly different clearance. See "Methods" for details. **c** Fitting of the initial slope to assign peptides to faster or slower clearance. **d** Example traces of phosphopeptides (yellow) whose clearance significantly deviates from the protein median (gray). Numbers over

facets represent phosphosites phosphorylated on the peptides in question. **e** Volcano-like plot showing peptides with significantly slower or faster clearance compared to the protein median. x axis shows the difference in fit compared to the protein median signed by the difference in initial slope: sign(Δinitial slope) *√(RSS$_0$−RSS$_1$). **f** Significantly more peptides in the phospho fraction exhibit differing clearance, with the majority of hits being slower. In contrast, most hits of unmodified peptides exhibit faster clearance. **g** Phosphoproteins are more likely to show differing clearance in at least one of their peptides compared to all proteins. **h** Phosphosites exhibiting faster clearance are more likely to be conserved and functionally important as shown by lower SIFTS scores[14,15]. Below: sample sizes of each group (*n*); above: *p* values from a two-sided Wilcoxon signed-rank test. Boxplots in **a** and **h** consist of median line, box: upper and lower quartiles, whiskers: 1.5 times interquartile range, points: outliers. Source data are provided as a Source data file.

proteoforms. The measured difference was not due to biases introduced in phosphoenrichment or measurement, as demonstrated by high correlation of clearance for peptides quantified in both the eluate and the total fraction (Supplementary Fig. 2c).

To estimate clearance of peptides, we first corrected the raw SILAC data for dilution due to cell division (see "Methods" for details) giving estimates of the fraction of old protein remaining (*φ*). We identified peptides with differing clearance using a comparative fitting approach comparing each peptide (unmodified or phosphorylated, representing *P*$_u$ and *P*$_P$, respectively) to the median of all (other) unmodified peptides of that protein, (representing the entire protein pool, *P*) (Fig. 2b, Supplementary Data 1, see also "Methods"). To generate high-confidence hits, we only included peptides measured in at least two replicates, four time points, and with a minimum of two unique unmodified peptides to constitute the protein median reference. We also excluded the last (28 h) time point due to it being temporally far removed from the rest of the timeseries. This resulted in statistical comparison for 2978 phosphopeptides (covering 999 gene names), and 29,286 unmodified

peptides (3573 gene names, see Supplementary Data 2 and Supplementary Data 3). Hits were classified into "faster" or "slower" clearance based on the initial slope (≤6 h) of the clearance profile representing an initial clearance rate, *k*$_{app}$ (Fig. 2c, d). In accordance with the larger overall spread of clearance of phosphorylated peptides, we found a significantly greater fraction of hits in phosphorylated than unmodified peptides (Fig. 2e, f), with 45% of all quantified phosphoproteins exhibiting at least a single peptide with significantly altered clearance (Fig. 2g). Hits were evenly distributed across phosphorylated amino acid residues (S, T, or Y, Supplementary Fig. 2d) and protein abundance in both sample types (Supplementary Fig. 2e).

It should be noted that, as a whole, our bottom-up approach is likely to underestimate differences in clearance of proteoforms, as measurements of peptides shared between multiple proteoforms represent an abundance-weighted mean over all proteoforms present (our *P*). Thankfully, however, this should only decrease the likelihood of false positives and increase the confidence of identified hits to actually represent distinct proteoform pools.

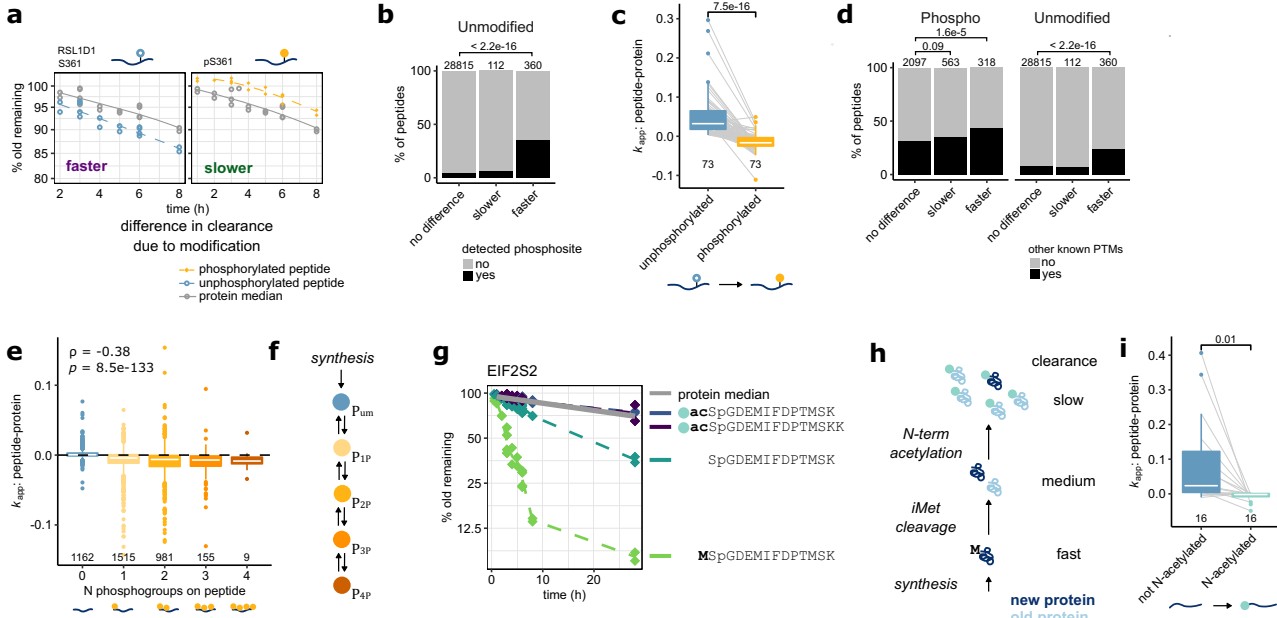

**Fig. 3 | Peptide clearance rates reveal intermediates of protein modification.**
**a** Change in clearance of a RSL1D1 peptide caused by phosphorylation of S361. Left: faster clearance of a RSL1D1 peptide containing unmodified S361 (blue, dashed line) compared to the median of the rest of the unmodified RSL1D1 peptides (gray, solid line). Right: slower clearance of a RSL1D1 peptide phosphorylated at S361 (yellow, dashed line). **b** Unmodified peptides with faster clearance are strongly enriched for the same peptide also detected in its phosphorylated form. Fisher's exact test, two-sided. **c** Initial clearance ($k_{app}$) of peptides carrying phosphosites, where the unmodified form was identified as showing faster clearance and a phosphorylated form was also quantified. Shown are median values over replicates and peptides in case the same phosphosite was detected on multiple peptides. Paired, two-sided $t$ test, ($t$-value = 10.12, degrees of freedom (df) = 72). **d** Phosphorylated peptides with faster clearance, and unmodified peptides with faster clearance are enriched for peptides carrying other known PTM sites (from Uniprot) suggesting that they represent early intermediates of protein modification cascades. Fisher's exact test,

two-sided. **e** Multiply-phosphorylated peptides trend towards lower clearance as number of phosphogroups increases. Peptides with zero phosphogroups are unmodified carrying residues, which were also detected in a phosphorylated form. Shown are peptides with slower or not significantly different clearance. ρ, Spearman correlation; $p$, associated $p$ value. These data suggest gradual, cumulative phosphorylation after synthesis as shown in (**f**). **g** Maturation intermediate peptides on the EIF2S2 N-terminus show faster clearance consistent with sequential, stepwise protein maturation depicted in the cartoon (**h**). **i** Non-acetylated N-terminal peptides from proteins undergoing N-acetylation have faster clearance consistent with them being maturation intermediates. Note that clearance of the final N-terminally acetylated form (in turquoise) closely follows the clearance of the protein median. $k_{app}$, initial slope of the clearance profile, see Fig. 2c. Paired, two-sided $t$ test ($t = 2.94$, df = 15). Boxplots in **c**, **e**, and **i** consist of median line, box: upper and lower quartiles, whiskers: 1.5 times interquartile range, points: outliers. Source data are provided as a Source data file.

## Slower clearance of phosphopeptides is the expected default behavior

In line with our prediction from the simplest PTM-containing model (Model 1:2, Fig. 1d), we find the majority (64%) of phosphopeptide hits exhibiting slower clearance (Fig. 2f). Importantly, as demonstrated by Model 1:2, this is most readily explained by slow addition of phosphate groups (i.e., low $k_w$). As this can result from numerous biological processes (including potential "non-functional" protein phosphorylation[13]), we deem these differences in clearance rate unlikely to be functionally informative when considered in isolation. This is also reflected in an overall younger evolutionary age of phosphosites with slower clearance (Supplementary Fig. 2f), often equated with lower likelihood of functional importance[14].

Conversely, however, peptides with faster clearance demand further explanation and likely represent a functionally more interesting subgroup than the slower hits. This is also represented by lower SIFTS scores (Structure Integration with Function, Taxonomy, and Sequences), which correspond to higher conservedness and higher predicted functional importance (Fig. 2h)[14,15].

## Faster clearance rates reveal protein maturation intermediates

In contrast to the slower clearance for the majority of phosphorylated peptides, we find that our unmodified hits are strongly enriched for faster clearance (76% of hits). Theory suggests that these peptides correspond to newly-synthesized or intermediate proteoforms, destined to undergo conversion to another form during protein

maturation or aging (Model 1:2 and 1:2r, Fig. 1d, e). Indeed, this is what we observe, as unphosphorylated peptides with faster clearance are strongly enriched for peptides, where a corresponding phosphopeptide was also measured (Fig. 3a, b). Furthermore, the corresponding phosphopeptide showed predominantly slower clearance (Fig. 3c) suggesting directionality of the modification with regard to protein maturation in line with Model 1:2 (Fig. 1d). Similarly, peptides with faster clearance are also enriched for sites of other previously-identified PTMs[16] (Fig. 3d, Supplementary Fig. 4a). The same trend was observed for multiply-phosphorylated peptides, where the number of accumulated phosphate groups correlated with slower clearance (Fig. 3e, Supplementary Fig. 2g), as expected from a step-wise modification cascade (Fig. 3f).

Strikingly, the effect that sequential, directed modification has on measured clearance is perhaps most clearly observable for N-terminal protein acetylation, a well-understood process thought to be irreversible in cells[17]. We measured multiple N-terminal peptides of Eukaryotic Translation Initiation Factor 2 Subunit Beta (EIF2S2), whose relative clearance rates are concordant with stepwise, sequential maturation (Fig. 3g). The species with the fastest clearance still contains the initiator methionine (light green in Fig. 3g), which, once removed by N-terminal methionine excision, creates the second intermediate species (dark green in Fig. 3g). Finally, maturation is completed with N-terminal acetylation (N-Ac) of the resulting peptide, whose clearance finally closely follows the median of all other EIF2S2 peptides, corresponding to the bulk of the fully mature EIF2S2 pool

(Fig. 3g, h). In total, we found 16 examples of N-terminally acetylated proteins, where both an acetylated and a non-acetylated N-terminal peptide could be quantified (Fig. 3i; Supplementary Fig. 2h, i), all of which showed relative differences in clearance in line with our Model 1:2 for irreversible protein modification (Supplementary Fig. 2j).

In addition, our high-resolution dataset also includes more typical proteoform-level effects visible as differing clearance rates of a subset of peptides. These include, for example, isoform-specific turnover of Nuclear autoantigenic sperm protein (NASP), signs of autoproteolytic cleavage of Nucleoporin 98/96 precursor[18], and partial degradation of Nuclear Factor Kappa B Subunit 2 (NFKB2)[19] (Supplementary Fig. 3). However, these represent the minority of the peptide hits detected, and will thus not be discussed further in the context of this study.

### Fast clearance due to concerted dephosphorylation

The eukaryotic cell cycle is known to give rise to pervasive, synchronized, and ordered changes in both protein phosphorylation[20,21] and synthesis[22,23]. Notably, this synchronization effect can be expected to remain visible in our PPToP data despite the use of asynchronous cultures, if phosphorylation and/or dephosphorylation occurs in a defined sequence with respect to a protein's synthesis. A prime example of this is the proliferation marker protein Ki-67 (MKI67), for which we observe wide-spread faster clearance of phosphorylated peptides (Fig. 4a, Supplementary Fig. 4b). Previous proteomic data[22] suggest that MKI67 is primarily synthesized in G2 phase coinciding with high MKI67 phosphorylation (Fig. 4b). Our data suggest that synthesis and phosphorylation is followed by an event of concerted MKI67 dephosphorylation. This could also explain the recently-described function of MKI67 in segregating premitotic chromosomes, which is followed by a molecular change in MKI67 (likely the dephosphorylation seen in our data) leading to chromosome condensation[24].

However, while proteins exhibiting cell cycle-dependent synthesis and/or phosphorylation patterns, such as MKI67, are slightly enriched in our PPToP dataset (Fig. 4c), they are unlikely to be the main contributing factor for the majority of hits found.

### Fast clearance is not predictive of low proteolytic stability in cells

Previous reports measuring turnover of PTM-modified peptides have focused on their potential effects on proteolytic stability, largely equating high measured clearance rates with fast protein degradation[6,7], and interpreting PTM sites with fast clearance as potential degrons. In contrast, our theoretical considerations predict that measured changes in clearance can be expected to be mostly unrelated to actual protein stability effects. We thus set out to test this prediction experimentally. We chose a diverse and representative set of 23 proteins carrying 65 phosphosites of interest (Supplementary Fig. 5a) covering a wide range of protein half-lives. 53 out of the 65 chosen phosphosites showed fast clearance in our primary PPToP screen (Supplementary Fig. 5a). We mutated each group of phosphosites to alanines (ALA, creating a phosphorylation-incompatible mutant) and/or aspartates (ASP, mimicking the negative charge of phosphoryl groups), and expressed them as mEGFP-fusions in HeLa cells (Fig. 5a and Supplementary Data 4) at roughly comparable expression levels (Supplementary Fig. 6a). We also used fluorescence microscopy to verify that mEGFP-fusion constructs showed expected physiological localization (Supplementary Fig. 5b), and discarded constructs with too low expression (1 discarded) or unphysiological localization (2 discarded).

If the measured differences in clearance of peptides in PPToP were due to differing proteolytic stability, one would expect the degradation rates of ASP mutants to correlate with the measured clearance rates of phosphopeptides, while ALA mutants should show no correlation or even anti-correlate. We thus quantified their effects on the protein's degradation rate by combining isobaric labeling and pulsed SILAC

(Fig. 5a). Fusion proteins were pulled-down using anti-GFP beads under harsh buffer conditions to isolate them from endogenous proteins before digest. Comparing degradation rates of each mutant to the corresponding wild-type, we found that while most mutations (both ASP and ALA) slightly lowered degradation (i.e., slightly stabilizing the protein, Supplementary Fig. 6b), these differences were not statistically significant for any of the sites tested (Fig. 5b). These data strongly suggest that differences in clearance measured by PPToP are independent of protein stability effects, and thus unlikely to directly represent degrons or stabilons. To further corroborate this, we compared PPToP hit phosphopeptides to predicted degron sequences[25], and found no enrichment in proximity to degrons for either faster or slower hits (Fig. 5c). Likewise, our PPToP hits were not proximal to any of 224 known human degron sequences[26].

### Maturation intermediate p-sites affect protein–protein interactions

As PPToP can detect protein maturation intermediates, we looked at protein complex assembly, where control of maturation is especially important as protein–protein interactions need to be established in a controlled manner, and unwanted interactions could be highly deleterious. Interestingly, our dataset includes phosphorylation sites with fast clearance on known protein complex subunits, representing prime candidates for maturation intermediates. In particular, we followed-up two phosphorylation sites: S16 and S56 on Proteasome subunit alpha type-5 (PSMA5). When phosphorylated, both sites exhibit faster clearance than their corresponding unphosphorylated peptides, which closely match the protein median (Fig. 6a), suggestive of transient early phosphorylation (Model 1:2r). Interestingly, both sites lie on protein–protein interfaces with neighboring subunits (PSMA1 and PSMA7) in the intact proteasome alpha ring and neither is phosphorylated in the mature complex (Fig. 6b). Pull-downs with wild-type (WT) PSMA5-GFP under stringent buffer conditions showed reproducible co-sedimentation of PSMA1, while neither the phosphodeficient (ALA) nor the phosphomimetic (ASP) mutant was able to pull down PSMA1 (Fig. 6c). Further pull-downs in less stringent buffer conditions confirmed that wild-type PSMA5-GFP interacts more strongly with PSMA1 than any of the mutants as well as showing significantly reduced co-sedimentation also of other proteasomal subunits in the mutant conditions (Supplementary Fig. 7a). In addition, previous profiling of thermal stability effects also showed phosphorylation of S16 and S56 to be associated with significantly lowered thermal stability of PSMA5 in HeLa cells[27] (Fig. 6d) as well as in yeast[28], suggesting that phosphorylated PSMA5 is not part of the assembled proteasome, as proteins assembled into larger complexes tend to show higher thermal stability[29,30]. This is further corroborated by subcellular fractionation experiments[31] that find PSMA5 phosphorylated on S16 only in the most soluble fraction, whereas unphosphorylated PSMA5 as well as other proteasomal subunits are also present in less soluble fractions (Supplementary Fig. 7b). Taken together, these data suggest that interaction with PSMA1 relies on functional phosphorylation of PSMA5 (most likely at S16), and that the phosphorylated species is not associated with the intact proteasome complex. Based on our PPToP data we suggest that phosphorylated PSMA5 represents a protein complex assembly intermediate potentially required for correct folding of PSMA5 (Fig. 6e).

We find that phosphorylated peptides with differing clearance are modestly, but significantly enriched in protein complex subunits also globally (Fig. 6f). Furthermore, comparing with earlier thermal proteomic profiling data[27], we find that differing clearance of phosphopeptides is also associated with changed thermal stability (Fig. 6g), again suggesting a distinct molecular state in cellulo for proteoforms defined by peptides identified by PPToP. We also find that peptides with faster clearance are enriched in intrinsically disordered regions of proteins (Fig. 6h), suggesting that these protein regions might act as

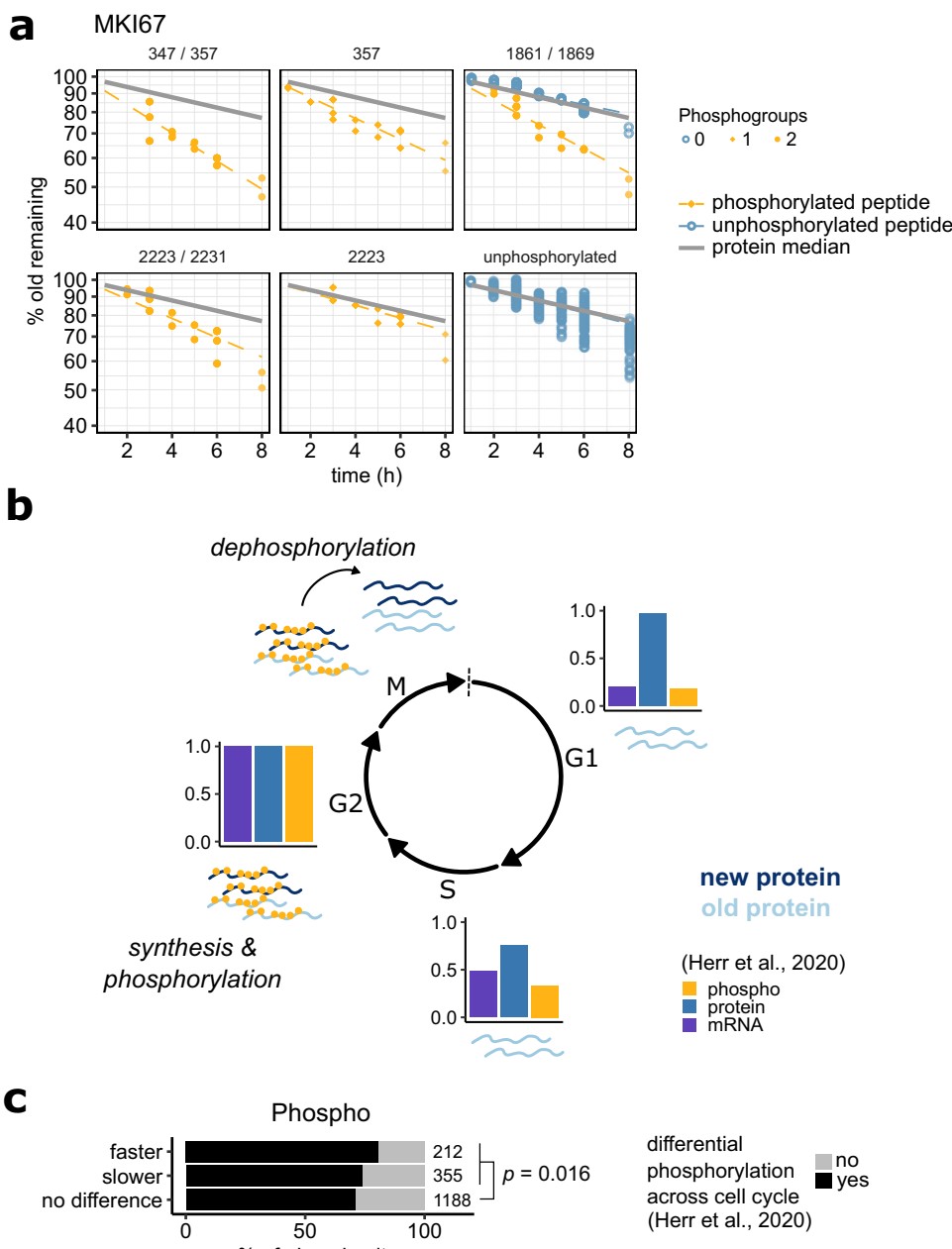

**Fig. 4 | PPToP captures cell-cycle-dependent effects. a** Representative examples of MKI67 phosphopeptides with faster clearance. Numbers above facets indicate phosphosites on the protein. Note that the peptide covering sites 1861 and 1869 was detected both phosphorylated (in yellow) and unphosphorylated (in blue) with distinct clearance profiles. **b** Our data combined with previous data profiling the phosphoproteome across the cell cycle[22] suggest that MKI67 is synthesized and phosphorylated in G2 phase and subsequently undergoes concerted

dephosphorylation in M phase. The dephosphorylation event suggested by the faster clearance measured by PPToP (**a**) explains previous reports of MKI67's role in condensation of chromosomes during M phase[24]. **c** Hits from the phospho fraction are enriched for peptides exhibiting cell cycle-dependent changes in phosphorylation or abundance[22]. Fisher's exact test, two-sided. Source data are provided as a Source data file.

specific switches between functionally distinct proteoforms, and that controlling the behavior of these regions (e.g., with PTMs) might be of special importance in the context of protein maturation. The enrichment in disordered regions in peptides with faster clearance was significant also after controlling for the confounding variable of other known PTMs, which tend to also be enriched in disordered regions (Supplementary Fig. 4c).

**Parameter estimation from PPToP data**
Given that our PPToP data was sampled with high temporal resolution and the resulting curves are information rich, we also

attempted fitting our theoretical models to the data. For this, we used a simplified Model 1:2 assuming that the two degradation constants $k_{deg,u}$ and $k_{deg,P}$ are equal (Fig. 7a). The reasons for this assumption were two-fold: firstly, the shape of the clearance curves is only weakly defined by the difference of $k_{deg,u}$ and $k_{deg,P}$ (Supplementary Fig. 1), and secondly, the decreased number of parameters allows estimating parameters with higher certainty. We chose Model 1:2 as our data contained many examples of phosphopeptides with slow clearance along with the corresponding unmodified peptide with faster clearance. We also restricted fits to cases, where all possible model observables ($O_P$, $O_u$, and $O$, see

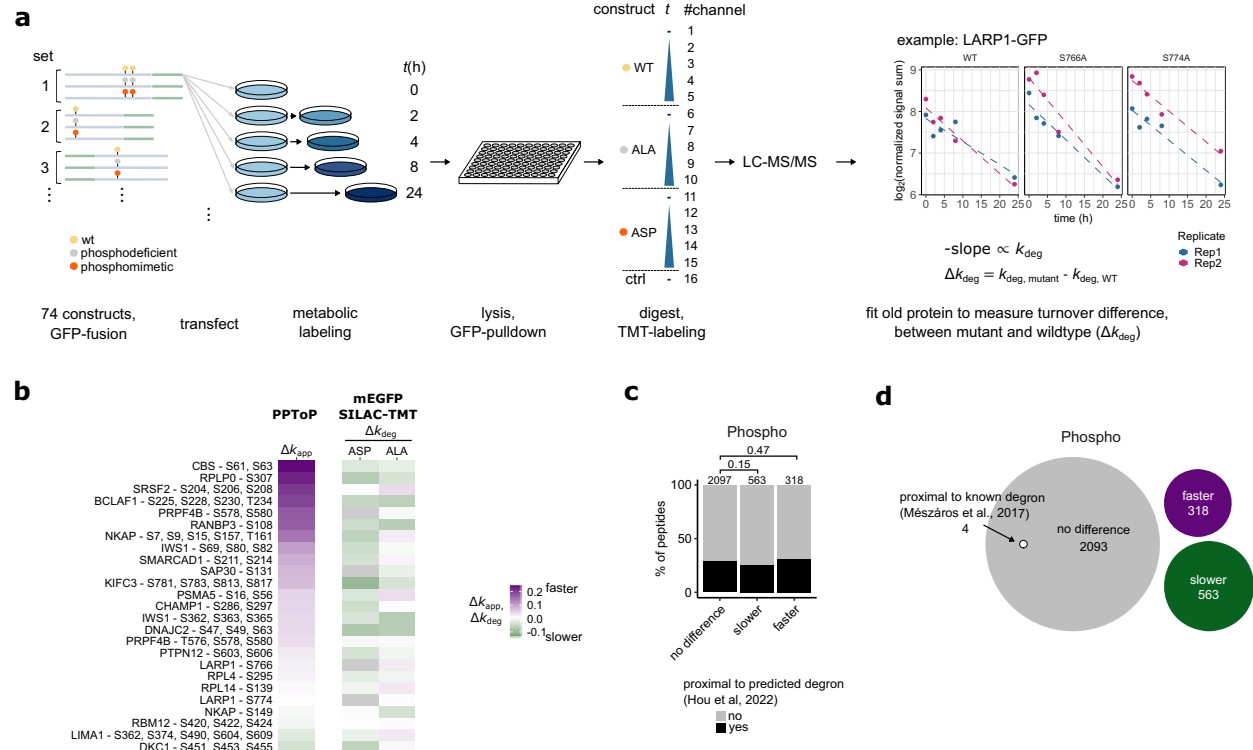

**Fig. 5 | Differential clearance of phosphopeptides is not predictive of effects of phosphorylation on a protein's proteolytic stability. a** Experimental setup for interrogating effects of phosphosite mutations on protein proteolytic stability. Turnover of exogenously expressed proteins was quantified from TMT signal sums of the old protein from pull-down experiments, and compared to the turnover of a corresponding, similarly-tagged and expressed wild-type (WT) construct. **b** Comparison of differences in clearance rates from PPToP ($\Delta k_{app}$) and wild-type-to-mutant differences ($\Delta k_{deg}$) from the exogenous expression experiment show no

significant correlation (see also Supplementary Fig. 6b), indicating that differences in clearance measured from PPToP are not predictive of differences in protein degradation caused by the PTM. **c** Phospho PPToP hits are not enriched for being proximal to predicted degrons[25]. Proximity was defined by considering whether any part of a predicted degron overlaps with a 31 amino acid stretch centered on the measured peptide. Fisher's exact test, two-sided. **d** No known degrons[26] lie proximal to the measured Phospho PPToP hits. Source data are provided as a Source data file.

Supplementary Note 1 Eqs. 42, 44, 67) were measured. We found that fitting could faithfully capture differences in the PPToP clearance curves, which manifests in differences in the rate constants for writing and erasing ($k_w$ and $k_e$, respectively; Fig. 7b). Comparing data from all N-Ac and phosphorylation sites fitted with high confidence (see "Methods" for details) showed distinct differences in the rate constants, with N-Ac exhibiting significantly faster writing (medians of theoretical mean times to modify 2.7 h for N-Ac, and 5.4 h for Phospho) and slower erasing, while the underlying protein stability ($k_{deg}$) showed no significant difference (Fig. 7c, d). The steady-state occupancy of a PTM site equals the fraction of modified protein in the entire protein pool in Model 1:2. This fraction is a function of $k_w$, $k_e$, and $k_{deg}$ (see Supplementary Note 1 4.2, Eq. 47), and can thus be calculated from the estimated parameter values. Interestingly, median occupancy of N-Ac sites was significantly higher than for Phospho (0.94 and 0.49, respectively, Fig. 7e). These data are well in line with the expected rapidity and irreversibility of N-Ac in cells[17]. Furthermore, the estimated occupancies of phosphorylation agree exceptionally well with experimentally estimated values from measurements comparing phosphatase-treated samples with untreated controls[32] (Fig. 7f).

## Discussion
PTMs, such as reversible phosphorylation, control all aspects of protein function from protein–protein interactions and catalytic activity to subcellular localization and proteolytic stability. Despite advances in the identification and localization of PTMs onto proteins by MS-based omics technologies, functional annotation and understanding

of PTMs is severely lagging behind as, e.g., >95% of known human protein phosphosites lack any annotation on biological function[33]. This lack of understanding of PTMs is especially pronounced in the temporal dimension. Specifically, how (or whether) PTMs are added at specific times over a protein's lifetime from its synthesis on a ribosome, folding, maturation and function, through to its eventual degradation, has so far remained largely elusive.

Based on experimentally validated theoretical considerations, we show here that peptide-level turnover analysis, such as PPToP, can be used to deliver exactly this information. Analogously to isotopic labeling in metabolic flux experiments[34], combining SILAC with MS effectively reveals the relative temporal order of events through observation of label incorporation along a network. Similarly, while we focus mainly on proteoforms defined by the addition and removal of PTMs, the same considerations and conclusions apply to any analogous metabolic labeling experiment of biomolecules in which a species can exist in multiple measurable states that can interconvert, such as nucleic acids and their modifications[35].

The information provided by PPToP allows establishing temporal ordering of events along the protein's lifetime. It thus complements previous findings, which have estimated that for around 10% of human proteins degradation rates are dependent on the age of the protein itself[8]. Our findings extend this notion by not only providing the opportunity to generate hypotheses about the PTMs involved in age-dependent stabilization, but also showing how wide-spread protein age-dependent modification is in the human proteome.

Using PPToP we identify numerous phosphosites of interest. These include sites on PSMA5, which we think could correspond to

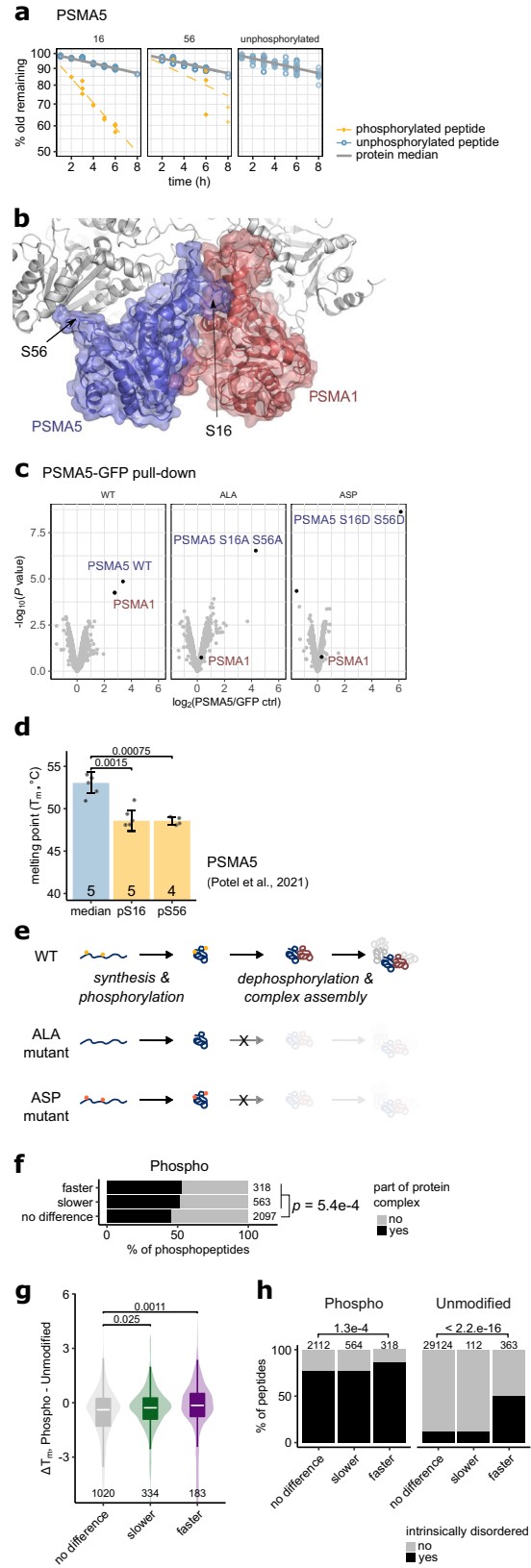

**Fig. 6 | Transient phosphorylation at S16 on PSMA5 is needed for interaction with PSMA1. a** Clearance profiles of PSMA5 peptides carrying S16 or S56 (yellow), as well as all unphosphorylated peptides (blue) are shown. **b** Structure of PSMA5 (in blue) in the mature proteasome. The neighboring alpha ring subunit PSMA1 is highlighted in red. S16 and S56 are highlighted as sticks. Both residues are unphosphorylated in the mature proteasome structure (PDB ID: 6MSB). **c** Pull-down under stringent buffer conditions (RIPA) of exogenously expressed PSMA5-GFP constructs in HeLa cells shows that the PSMA5-PSMA1 interaction is lost upon mutation of S16 and S56. y axis: unadjusted $p$ value from a limma analysis. $N = 2$. See also Supplementary Fig. 7a. **d** Thermal stability of peptides phosphorylated at S16 and S56 is significantly lower than the PSMA5 median suggesting a different biophysical state for phosphorylated PSMA5. Data from ref. [27]. $t$ test (median-pS16: $t = 4.75$, df = 8.00; median-pS56: $t$: 6.88, df = 5.36). Error bars are SD. Number of individual replicates shown below. **e** Hypothesis for the role of S16 (and S56) phosphorylation. We hypothesize that transient S16 and S56 phosphorylation is required for PSMA5 maturation and its incorporation into the proteasome. **f** Phosphopeptides exhibiting differing clearance are enriched in protein complex subunits. Fisher's exact test, two-sided. Protein complexes are from the CORUM core complex dataset[47]. **g** Phosphopeptides with differing clearance exhibit altered thermal stability compared to the unmodified protein median suggesting altered molecular states such as protein–protein interactions. Proteome-wide thermal stability data from ref. [27]. Two-sided Wilcoxon test. Boxplots consist of median line, box: upper and lower quartiles, whiskers: 1.5 times interquartile range. Outliers have been omitted for clarity. **h** Peptides with faster clearance are enriched in intrinsically-disordered protein regions. Disorder prediction from D2P2 (ref. [43]). See "Methods" for details. Fisher's exact test, two-sided. Source data are provided as a Source data file.

clearance, despite the peptide only being detected in two time points (see PSMB7 in the interactive data browser accessible at https://apps.embl.de/pptop).

Recently, two groups have published analyses of pSILAC data with the goal of achieving proteoform-resolution of protein turnover by estimating the turnover of single, proteoform-specific peptides[7], and this approach has been proposed to reveal the effects of PTMs on protein stability[6]. Using insight from our theoretical modeling, we show here that the majority of differences in measured clearance rates are not linked to protein proteolytic stability differences, but rather are indicative of the rate of PTM addition and network wiring. Results by Zecha et al.[7] on lysine acetylation showing preferentially slower rates of clearance are thus suggestive of slow rates of addition, which is in line with the generally very low stoichiometry of lysine acetylation[37].

We also directly test the hypothesis that proteoform clearance rates are indicative of proteolytic stability. Using a substantial number of representative target proteins in our mutagenesis experiment, we found no positive correlation between differences in clearance rates from PPToP and differences in cellular degradation rates of phosphosite mutants and wild-type constructs (Fig. 5b, Supplementary Fig. 6b). In fact, we found a slight negative correlation for the phosphomimetic aspartate mutant (the opposite of what would be expected for actual phospho-degrons). But even though this was marginally statistically significant, the magnitude of the effect was very small (Supplementary Fig. 6b) and thus unlikely to be biologically significant. Based on both our theoretical predictions and our experimental validation, we thus conclude that proteoform clearance rates are not directly indicative of proteolytic stability effects.

It should be noted, however, that these conclusions do not in any way invalidate the use of pSILAC to measure protein degradation, when applied to an entire pool of proteoforms of a protein[1,2]. Rather, our conclusions highlight the need for more careful consideration, when measuring introduction of a metabolic label into a network of interconvertible species, such as proteins undergoing modifications by PTMs.

Interestingly, pSILAC analysis of ubiquitinated peptides has shown a propensity for higher rates of clearance for peptides carrying

proteasome assembly intermediates. Interestingly, we also observe effects of previously known proteasomal protein maturation events in our data. Multiple proteasomal β subunits undergo proteolytic processing during proteasome assembly[36], including PSMB7, which is cleaved after residue 43. Our PPToP data include a glimpse of this, as we detected a PSMB7 peptide starting at residue 42 with very fast

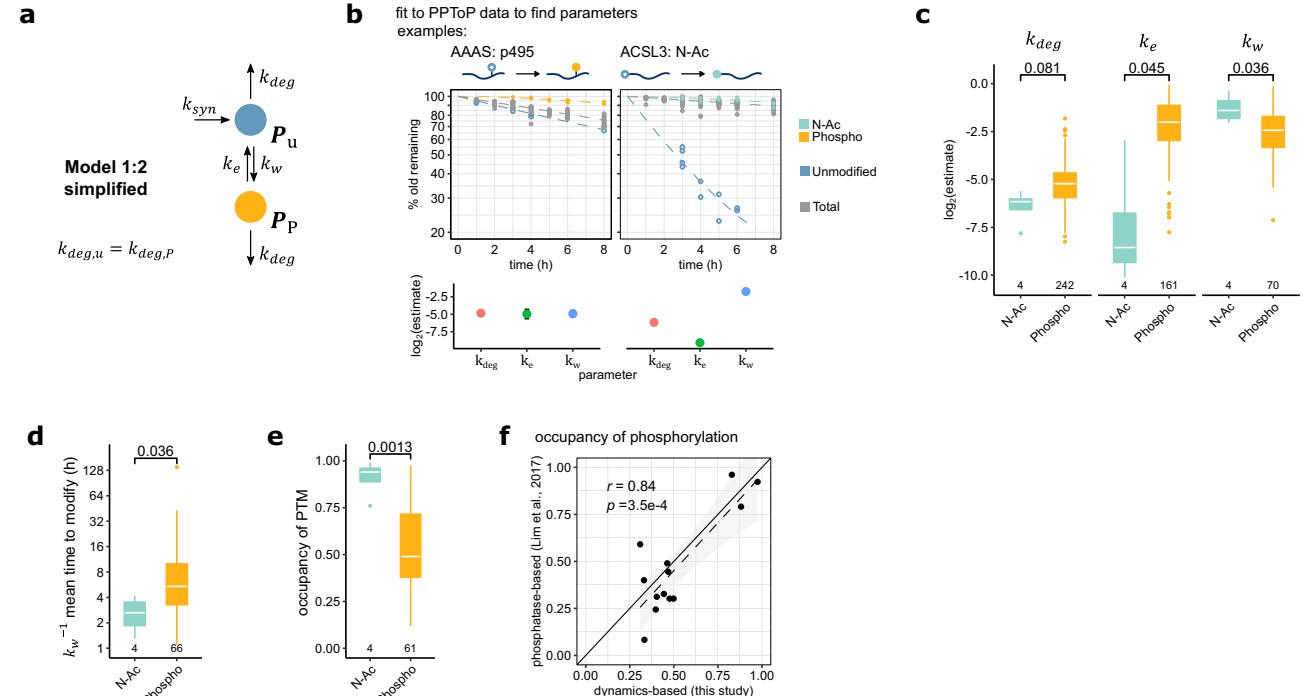

**Fig. 7 | Extracting parameters of PTM dynamics from PPToP data. a** Simplified Model 1:2 used for fitting. **b** Upper panel: example clearance traces and fits of two proteins. The N-Acylated or phosphorylated species are treated as the modified species $P_P$ in the fit. Lower panel: estimates for the different parameters for the proteins shown above. Shown are parameter estimates and standard deviations. **c** Writing and erasing rates, but not degradation are significantly different in N-Ac and phospho cases. Only parameters fitted with high confidence are shown (see "Methods" for details). Two-sided $t$ tests ($k_{deg}$, $t = -2.45$, df = 3.15; $k_e$, $t = -3.3$, df = 3.03; $k_w$, $t = 3.08$, df = 4.05). The conclusions were unaffected by the specific cutoff for confidence used. **d** Theoretical mean time to modify ($k_w^{-1}$) is significantly

shorter in N-Ac indicating faster modification. Two-sided $t$ test ($t = -3.06$, df = 4.16). **e** Steady-state occupancy of PTM (stoichiometry of modification) estimated from fitted parameters show significantly higher occupancy for N-Ac modified peptides. Two-sided $t$ test ($t = 6.17$, df = 5.28). **f** Occupancy of phosphorylation sites calculated from PPToP dynamics-based parameters agree well with experimental estimates. Pearson correlation. $p$ value for null hypothesis testing, i.e., no correlation (two-sided). The shaded region indicates the 95% confidence interval of the fit. Boxplots in **c**, **d**, and **e** consist of median line, box: upper and lower quartiles, whiskers: 1.5 times interquartile range, points: outliers. Source data are provided as a Source data file.

ubiquitin-remnant sites[7]. This is noteworthy, as it suggests that these ubiquitination events occur preferentially on relatively recently-synthesized proteins, compared to the rest of the protein's proteo-forms. Faster clearance rates were especially enriched for ubiquiti-nated ribosomal proteins, and we speculate that many of these identified ubiquitination sites might act as quality control switches during ribosome biogenesis, potentially marking misfolded or mis-incorporated subunits that might be subsequently degraded. This behavior is in line, e.g., with our Model 1:3 (Supplementary Fig. 8), where fast clearance for the modified pool can be combined with rapid degradation in the case of an early bifurcation of protein fate (Supplementary Fig. 8b: "early PTM degron", also accessible through the interactive model output browser at https://apps.embl.de/pptop). Notably, however, the same behavior can also be explained by deubi-quitination during protein maturation, which would likewise lead to faster clearance rates (Supplementary Fig. 8b: "maturation inter-mediate PTM site"). More targeted experiments, including mutagen-esis of the identified sites, as well as careful examination of the clearance profile of the entire protein pool ($P$)[8] will be needed to dis-tinguish between these two potential explanations.

In summary, we found that peptides exhibiting faster clearance are enriched, and thus are likely explained by at least the following mechanisms: other PTMs (leading to further modification and con-current clearance of the original peptide), concerted synthesis and (de)phosphorylation along the cell cycle, and protein maturation during complex formation. Interestingly, however, a large proportion of peptides with faster clearance currently remain without immediate mechanistic explanation (Supplementary Fig. 9). We think these

previously unannotated sites thus present an exciting source for future research.

## Methods

### Cell culture and isotopic labeling

HeLa Kyoto cells (from S. Narumiya, RRID: CVCL_1922, see also Sup-plementary Data 5) were cultured according to standard tissue culture techniques in SILAC DMEM Flex Media (Gibco) containing either "Light" ($^{12}$C and $^{14}$N-labeled, Arg0 and Lys0, Thermo) or "Heavy" ($^{13}$C and $^{15}$N-labeled, Arg10 and Lys8, Silantes) labeled arginine and lysine, 1 mg ml$^{-1}$ glucose, 10% dialyzed FBS (Gibco), and 1 mM L-glutamine. Cells were grown for a minimum of 10 doublings in the respective media before starting a time course to ensure complete labeling. For the SILAC time course 2.3E6 cells were seeded onto 15 cm dishes in Light (replicates 1 through 3) or Heavy medium (replicate 4) and cul-tured for 24 h to 48 h before medium exchange. A single 15 cm dish was used per time point per replicate. Medium was aspirated and cells washed twice with warm PBS (with Ca and Mg), then medium replaced with Heavy (replicates 1 through 3) or Light medium (replicate 4). For each replicate all dishes were seeded and lysed simultaneously with staggered medium exchange.

Both cell lines used were verified to be mycoplasma free.

### Sample preparation and phosphoenrichment

Cells were washed twice with ice-cold PBS and scraped on ice into lysis buffer (6 M Urea, 100 mM Hepes (pH 8.5), 5 mM Tris(2-carboxyethyl) phosphine (TCEP), 30 mM chloroacetamide, 4 mM MgCl$_2$, 2 mM NaVO$_4$, 2 mM NaF, 2 mM Na-pyrophosphate, 1% sodium deoxycholate,

1% Triton X-100, 1x Complete protease inhibitor cocktail (Roche), 1x PhosStop phosphatase inhibitor cocktail (Roche), 0.5% Benzonase (Merck, 70746)), diluted 1:1 with dilution buffer (6 M Urea, 100 mM Hepes (pH 8.5), 4 mM $MgCl_2$), and sonicated at +4 °C in a Bioruptor Plus, 45 cycles, 30 s on/30 s off (Diagenode). Lysates were cleared by centrifugation, $16,000 \times g$, at 4 °C for 60 min and supernatants frozen to −80 °C before continuing. Cleared supernatants were thawed at room temperature (RT) and nucleic acids digested by adding 0.5% fresh Benzonase and incubating 1 h at RT, after which EDTA was raised to 25 mM and SDS to 2%. Protein was precipitated in a 4:4:1 lysate:methanol:chloroform mixture, centrifuged 10 min at $4000 \times g$, RT, and the resulting protein precipitate extracted and washed twice with 70% EtOH in a sonicator water bath. Protein was resuspended to 2.5 mg ml$^{-1}$ into digestion buffer (100 mM Hepes (pH 8.5), 5 mM TCEP, 30 mM chloroacetamide, 1% sodium deoxycholate), TPCK-Trypsin (Thermo, 20233) added to 100 µg ml$^{-1}$ and incubated on an end-over-end shaker overnight at RT. Digested peptides were desalted on SepPak columns using gravity flow (Waters, WAT054945), washed twice with 0.1% trifluoroacetic acid (TFA), eluted with 40% acetonitrile (ACN), and resulting peptides lyophilized.

Phosphopeptides were enriched using a ProPac Immobilized Metal Ion Affinity Chromatography (IMAC)-10 column (Thermo, 063276) loaded with $Fe^{3+}$ on a Dionex Ultimate 3000 HPLC system (Thermo Fisher Scientific)[27,38] as follows: lyophilized peptides were resuspended in buffer A (70% ACN, 0.07% TFA) and injected onto the column at a flow rate of 400 µl min$^{-1}$ for 6 min. Subsequently, the column was washed with 100% buffer A for 6 min at 1 ml min$^{-1}$, and phosphopeptides eluted with 50% buffer B (0.3% ammonia) for 2 min at a flow rate of 500 µl min$^{-1}$. The phosphopeptide-containing eluate as well as the flow-through were collected and lyophilized.

## High-pH peptide prefractionation

5% of the flow-through ("total") was taken up in 20 mM ammonium formate (pH 10) and prefractionated first into 29 fractions on a 1200 Infinity HPLC (Agilent) using high-pH reversed-phase chromatography (running buffer A: 20 mM ammonium formate pH 10; elution buffer B: ACN) on an X-bridge column (2.1 × 10 mm, C18, 3.5 µm, Waters). Fractions were then pooled across to generate 12 fractions, and vacuum dried.

Phosphopeptides were fractionated manually using in-house packed C18 microcolumns[27] into seven fractions as follows: gel-loader tips were plugged with C18 resin (Affinisep AttractSPE C18 Disks) and packed with approximately 1 mg C18 material (Dr. Maisch, 5 µm, 120 Å). Lyophilized phosphopeptides from enrichment were taken up using 40 µl buffer A (20 mM ammonium formate at pH 10) and loaded into the microcolumn by centrifugation using speeds resulting in approximate loading speeds of 10 µl min$^{-1}$. Columns were washed with 10 µl buffer A, and both the flow-through and wash collected as a first fraction (FT). Subsequently, phosphopeptides were fractionated using a stepped gradient with sequential addition and elution using 10 µl of the following solutions: 1%, 3%, 5%, 7%, 9%, 11%, 13%, 15%, 17%, 19%, 21%, 23%, 24%, 26%, 28%, 30%, 35%, and 40% ACN in buffer A. Elution was done using centrifugation matching flow rates to approximately 10 µl min$^{-1}$. Elutions were cross-pooled as follows: F1: 1%, 13%, 24% ACN; F2: 3%, 15%, 26% ACN; F3: 5%, 17%, 28% ACN; F4: 7%, 19%, 30% ACN; F5: 9%, 21% 40% ACN; F6: 11%, 23% ACN.

## Experiments with exogenously expressed GFP-fusion constructs

23 proteins identified as carrying phosphorylation sites of interest from the proteome-wide experiment were chosen for follow-up studies. Proteins were chosen as representative examples including proteins with diverse whole-protein half-lives and varying levels of prior published information. For this, Pubmed and PhosphoSitePlus were accessed on 16.11.21 and searched using the gene name or common alternative names, where applicable (Fig. 5a). Genes encoding proteins

of interest were cloned as either N or C terminal fusions to monomeric EGFP (mEGFP, carrying the A206K mutation) in a vector under a Ubc promoter (kind gift from Daniel Heid and Judith Zaugg; similar to Addgene plasmid 11155). Phosphorylation sites of interest were mutated to either alanine and/or aspartate (see Supplementary Data 4 for full list of constructs). Cloning and mutagenesis was performed by GenScript (GenScript Biotech Netherlands B.V.).

## Microscopy

To ascertain correct subcellular localization, GFP-fusion constructs were expressed in HeLa cells as follows: 4000 cells per well in 150 µl medium were seeded onto glass-bottom 96-well plates (Greiner) the day prior to transfection. Cells were transfected using FuGENE6 (Promega) according to manufacturer's instructions using 0.125 µl FuGENE reagent and 50 ng plasmid per well. Cells were fixed 72 h after transfection using 4% paraformaldehyde (Pierce) in PBS for 30 min, washed thrice with PBS, permeabilized for 15 min using 0.05% Triton X-100 in PBS, stained with a mixture of Hoechst 33342 (Biotrend, 1:5000) and Phalloidin-Atto647N (Sigma, 1:1000) in PBS for 1 h, blocked using 10% Normal Goat Serum (Thermo) for 30 min, stained using anti-GFP (Abcam, ab1218, 1:500 in 1% BSA, 5% Normal Goat Serum, 0.01% Tween 20 in PBS) for 1 h, washed thrice with 0.1% Tween in PBS (PBST), then stained with Alexa 488 anti-mouse (Invitrogen A11001, 1:2000 in same buffer as anti-GFP above) for 1 h, washed again thrice in with PBST, and stored in PBS. Cells were imaged on a Nikon eclipse Ti automated microscope using a Plan Apo λ 20x objective in widefield mode.

Subcellular localizations of exogenously expressed proteins were compared to the reference localization from the Human Protein Atlas (https://www.proteinatlas.org/)[39], accessed 26.7.2022, or (when not available) to Uniprot. Only constructs with correct localization were chosen for further experiments. One protein target was discarded due to unphysiological localization.

## AP-MS: pull-down and sample preparation

Constructs were expressed in HeLa cells as follows. Cells were seeded onto 12-well plates at 90E3 cells per well in Light SILAC medium the day prior to transfection using a single well per construct, per time point, per replicate. Each well was transfected with 360 ng of plasmid using Lipofectamine 3000 (Invitrogen) according to manufacturer's instructions, and let to transfect for 24 h, after which medium was exchanged to Heavy medium for all wells simultaneously. Cells were collected after the designated time in Heavy medium, placed on ice, washed with PBS and lysed in RIPA lysis buffer (50 mM Hepes pH 7.5, 150 mM NaCl, 0.5 mM EDTA, 0.1% SDS, 1% Triton X-100, 1% sodium deoxycholate), NP40 lysis buffer (50 mM Hepes pH 7.5, 150 mM NaCl, 0.5 mM EDTA, 1.6% NP40), or detergent-free Freeze-thaw lysis buffer (50 mM Hepes pH 7.5, 150 mM NaCl, 1.5 mM $MgCl_2$), all with 1x Complete protease inhibitor cocktail (Roche), 2 mM NaF, 2 mM NaPP (sodium pyrophosphate), and 2 mM $NaVO_4$. After lysis, cells were collected by scraping, diluted 1:1 with dilution buffer (50 mM Hepes pH 7.5, 150 mM NaCl, 5 mM $MgCl_2$, 0.625 U µl$^{-1}$ Benzonase (Merck, 70746)), and left on ice for >30 min. For detergent-free lysis ("Freeze-thaw"), scraped cells in solution were vortexed briefly, then snap-frozen in liquid $N_2$ and thawed at RT for 5 min thrice. Lysates were cleared by spinning 5 min at $2000 \times g$ at 4 °C, and filtering through a pre-wetted 0.22 µm filter plate (Merck Millipore). For the GFP pull-down, cleared lysate was incubated with 2 µl washed GFP-Trap magnetic agarose beads (Chromotek) per well for >4 h at 4 °C, washed twice with wash buffer with NP40 (50 mM Hepes pH 7.5, 150 mM NaCl, 5 mM $MgCl_2$, 0.05% NP40), and thrice with detergent-free wash buffer (50 mM Hepes pH 7.5, 150 mM NaCl, 5 mM $MgCl_2$). Proteins were digested on beads with trypsin and Lys-C (5 ng/µl final concentration each) in 90 mM HEPES (pH 8.5), 5 mM chloroacetic acid and 1.25 mM TCEP overnight at RT shaking at 500 rpm. Peptides were eluted using 2% DMSO and dried in a speedvac.

Dry peptides were reconstituted in 5 μl water and labeled by adding 2 μl TMT label (20 μg μl$^{-1}$ in acetonitrile (ACN)) (TMTpro 16 plex, Thermo Fisher Scientific) and incubating 1 h at RT. Labeling was quenched with hydroxylamine (1.1% final concentration), and samples pooled to make full TMT16 sets as shown in Fig. 5b. Pooled sets were desalted on an OASIS HLB μElution plate (Waters 186001828BA); washing thrice with 0.05% FA, eluting with 80% ACN, 0.05% FA, and drying in a speedvac.

### Mass spectrometry

**Proteome-wide PPToP.** For LC-MS/MS analysis, peptides were reconstituted in 0.1% FA, 4% ACN and analyzed by nanoLC-MS/MS on an Ultimate 3000 RSLC (Thermo Fisher Scientific) connected to a Fusion Lumos Tribrid (Thermo Fisher Scientific) mass spectrometer, using an Acclaim C18 PepMap 100 trapping cartridge (5 μm, 300 μm i.d. × 5 mm, 100 Å) (Thermo Fisher Scientific) and a nanoEase M/Z HSS C18 T3 (100 Å, 1.8 μm, 75 μm × 250 mm) analytical column (Waters). Solvent A: aqueous 0.1% FA; Solvent B: 0.1% FA in ACN (all solvents LC-MS grade from Fisher Scientific). Instruments were controlled through Xcalibur (4.3) (Thermo Fisher Scientific).

LC-MS/MS analysis parameters for total proteome and phospho-fraction were as follows: Peptides were loaded on the trapping cartridge using solvent A for 4 min (3 min for phospho) with a flow of 30 μl min$^{-1}$. Peptides were separated on the analytical column at 40 °C with a constant flow of 0.3 μl min$^{-1}$ applying a 100 min gradient of 4–25% of solvent B in A, followed by a 5 min gradient (25–40%), and a 4 min washing step at 85% solvent B (both total and phospho). Peptides were directly analyzed in positive ion mode with a spray voltage of 2.4 kV and an ion transfer tube temperature of 275 °C (both total and phospho). Full scan MS spectra with a mass range of 300–1500 m/z (375–1500 m/z for phospho) were acquired on the orbitrap using a resolution of 120,000 (60,000 for phospho) with a maximum injection time of 50 ms (20 ms for phospho) and Normalized AGC Target of 50% (Standard for phospho). Data-dependent acquisition was used with a cycle time of 2 s (3 s for phospho). Precursors were isolated on the quadrupole with an intensity threshold of 1e3 (2e5 for phospho), charge state filter of 2–7, and an isolation window of 1.2 m/z (1.4 m/z for phospho). Precursors were fragmented using HCD at 30% (32% for phospho) collision energy. For the total proteome: MS/MS spectra were acquired on the ion trap, with a maximum injection time of 50 ms, and a dynamic exclusion window of 45 s. For the phospho-enriched fraction, MS/MS spectra were acquired on the orbitrap, at 30,000 resolution, a maximum injection time of 75 ms, and a dynamic exclusion window of 20 s.

**AP-MS for analysis of mEGFP-fusion proteins.** TMT16-labeled peptides from pull-down experiments were reconstituted in 0.1% FA, 4% ACN and analyzed by nanoLC-MS/MS on the same hardware described above. LC-MS/MS analysis parameters were as follows: Peptides were loaded on the trapping cartridge using solvent A for 3 min with a flow of 30 μl min$^{-1}$. Peptides were separated on the analytical column at 40 °C with a constant flow of 0.3 μl min$^{-1}$ applying a 104 min gradient of 6–28% of solvent B in A, followed by a 4 min gradient (28–40%), and a 4 min washing step at 80% solvent B. Peptides were directly analyzed in positive ion mode with a spray voltage of 2.2 kV and an ion transfer tube temperature of 275 °C.

Full scan MS spectra with a mass range of 375–1500 m/z were acquired on the orbitrap using a resolution of 120,000 with a maximum injection time of 50 ms. Data-dependent acquisition was used in top 10 mode. Precursors were isolated on the quadrupole with an intensity threshold of 2e5, charge state filter of 2–7, and an isolation window of 0.7 m/z. Precursors were fragmented using HCD at 34% collision energy. MS/MS spectra were acquired on the orbitrap, at 30,000 resolution, a maximum injection time of 100 ms, scan range in first mass mode (at 110 m/z), and a dynamic exclusion window of 20 s.

**Data analysis.** All data analysis was carried out in R (version 4.0.0 or later)[40]. All boxplots shown follow the same structure: line denotes median, box limits denote upper and lower quartiles, whiskers represent 1.5 times the interquartile range, and outliers outside of these are shown as points unless otherwise noted.

### Proteome-wide PPToP

**Analysis of MS raw files.** For PPToP MS raw files were processed using MaxQuant (version 1.6.4.0)[41] using a reference human proteome (uniprot Proteome ID: UP000005640, downloaded 9.6.2020). Data were processed separately for total and phospho-enriched samples, but for each all time points, fractions, and replicates were run together. Default search parameters were used, except as follows: multiplicity: 2; Heavy channel: Arg10, Lys8; variable modifications: Acetyl (Protein N-term), Oxidation (M), and only for the phosphofraction: Phospho (STY); fixed modifications: Carbamidomethyl (C); maximum number of modifications per peptide: 5; maximum missed cleavage sites: 2 (3 for phospho); LFQ: none; re-quantify: unchecked; match between run: checked.

**Data filtering and preprocessing.** Identified peptides from the Max-Quant evidence file were filtered to remove hits from the reverse database and potential contaminants. All subsequent analysis was done on the modified peptide level (henceforth referred to as "peptide") including information on Heavy amino acid incorporation, N-terminal acetylation (N-Ac), and phosphorylation, but excluding methionine oxidation. Peptides quantified in only one of the SILAC channels (constituting 45.3% of all identified peptides) were removed. In case a peptide was quantified multiple times, a single entry was chosen by choosing the species with (i) the lower posterior error probability (PEP), and (ii) the highest intensity. Peptides were further filtered for presence in at least 2 replicates and 2 time points.

Cell cycle times for each replicate were estimated from the protein median values in the unmodified fraction. Assuming exponential decay of most proteins, we have the linear relationship[1]:

$$ln\left(\frac{new}{old} + 1\right) = \left(k_{\text{deg}} + \frac{ln2}{t_{cc}}\right)t \qquad (1)$$

where $\frac{new}{old}$ is the SILAC ratio of new and old protein, $k_{\text{deg}}$ is the protein-specific degradation constant, and $t_{cc}$ the cell cycle time. We estimated $t_{cc}$ from the 1% longest-lived proteins where, assuming no active protein degradation ($k_{\text{deg}} = 0$), the slope is defined by $\frac{ln2}{t_{cc}}$, from which we got $t_{cc}$ estimates of 28.0 h, 26.5 h, 27.0 h, and 22.2 h for replicates 1 through 4, respectively. Subsequently, new/old SILAC ratios were transformed into fraction of old remaining (see Supplementary Note 1 for reasons of doing so), and corrected for cell cycle as follows:

$$corrected\ old\ remaining = \varphi = ln\left(\frac{new}{old} + 1\right)^{-1} + \frac{ln2}{t_{cc}}t \qquad (2)$$

Next, peptide entries were filtered for reproducibility between replicates by calculating the distance of the corrected old remaining (φ) value for each peptide for each replicate to the median value of that peptide at that time point over all replicates, and excluding entries with values deviating from the median by more than two standard deviations of the entire distance distribution of all peptides at all time points in that fraction (unmodified or phospho). This removed 2.3% of all measurements.

**Comparative fitting to find peptides deviating in clearance from the rest of the protein.** The clearance of each peptide was compared to the median of all other quantified peptides of that protein from the unmodified fraction by fitting a spline with 3 degrees of freedom to the trace of φ vs time excluding the last time point (28 h). Peptides with

data in at least 4 time points, a total of at least 6 data points, and for which the median of the other peptides in that protein included at least 2 unique peptides in the unmodified fraction were included. An *F*-statistic was calculated for fitting the spline to either both peptide and protein median together (H0 model) or to each separately (H1 model). Due to heteroscedasticity of the data, the resulting *F*-statistic was calibrated as delineated in ref. 42 by estimating the "effective degrees of freedom" ($d_1$, $d_2$) from a null dataset, where "peptide" or "median" labels were randomized (thus reducing any differences between the two classes to those occurring by chance). Since our dynamic PTM-SILAC dataset had differing amounts of data points per case, $d_1$, $d_2$ were estimated for six separate bins of the data depending on the number of data points available for the comparison. The thusly calibrated *F*-statistic distribution across both fractions was used to calculate *p* values for each peptide using the `pf()` function from the *stats* package in R and corrected for multiple testing using Benjamini–Hochberg correction. Cases with adjusted *p* value <= 0.001 were considered as hits.

**Matching phosphorylated and unphosphorylated peptides.** To compare behavior of peptides in their phosphorylated and unmodified form (e.g., Fig. 3b, c), phosphosites were collated on the site level, i.e., on the specific modified amino acid. Consequently, the specific peptide boundaries were disregarded, when matching unmodified and modified peptides as phosphorylation can cause miscleavages and thus change the tryptic peptide boundaries.

**Disordered protein prediction.** Prediction of protein disorder was taken from the D2P2 database[43] (https://d2p2.pro/) using a consensus threshold of 75% across the individual predictor algorithms when determining the disorder status per amino acid. For the enrichment analysis, a peptide was considered intrinsically disordered if it contained at least 40% disordered amino acids.

**Proximity to predicted or verified degrons.** Peptides were considered to be proximal to a degron sequence, if a stretch of +/−15 amino acids around the center of each peptide (31 amino acids altogether, unless the peptide was close to the N or C terminus) showed any overlap with a degron. Predicted degrons were taken from ref. 25 verified degrons from ref. 26.

**Other statistical analyses.** Statistical tests were done in R using the following functions: Fisher's exact test (Figs. 3b, 3d, 4c, 6f): `fisher.test()`; *t* test (Figs. 3c, 3i, 6d, 6h, Supplementary Fig. 2g) and Wilcoxon signed-rank test (Figs. 2h, 6g, Supplementary Fig. 2f): `t.test()` and `stat_compare_means()`; linear regression: `lm()`; Spearman correlation (Fig. 3e, Supplementary Fig. 6b): `cor.test()`; Pearson correlation (Fig. 7f): `stat_cor()`. Sample sizes are shown in each figure. Other covariates tested: on protein level: subcellular location, length; on peptide level: length, presence of signal or propeptides, intensity, non-tryptic cleavage, functional score[14], in interface with other proteins, motifs, secondary structure predictions, predicted phosphorylating kinase.

**Targeted analysis of exogenously expressed GFP-fusion proteins**
**Analysis of MS raw files.** MS raw files were processed using IsobarQuant[44] and peptide and protein identification was obtained with Mascot 2.5.1 (Matrix Science) using a reference human proteome (uniprot Proteome ID: UP000005640, downloaded 9.6.2020) modified to include the overexpressed protein constructs in question, known common contaminants and reversed protein sequences. Mascot searches were done only for old (light SILAC label) proteins. Parameters were: Trypsin/P; max. 3 missed cleavages; peptide tolerance 10 ppm; MS/MS tolerance 0.02 Da; fixed modifications: Carbamidomethyl (C), TMT16plex (K); variable modifications: Acetyl (Protein N-term), Oxidation (M), Phospho (ST), Phospho (Y), TMT16plex (N-term).

**Turnover analysis of exogenously expressed proteins.** Data from pull-down experiments were analyzed from the IsobarQuant peptide output file, filtering peptides to exclude contaminants (including skin and keratin contaminants), peptides without reporter (TMT) quantification data, peptides lacking K or R residues (e.g., C-terminal peptides), and peptides with FDRs > 0.01. In cases where a modified peptide was measured multiple times, the entries were collapsed to a single value choosing the peptide with the highest score and highest precursor-to-threshold (p2t) values. As the vast majority of peptides in all pull-downs were shared, and data was to be analyzed comparing each mutant construct to the respective, in-set WT sample, reporter signals were median-normalized over all channels.

Next, construct-specific peptides were identified and excluded from the analysis in channels, where the construct in question was not experimentally expressed (e.g., peptides carrying an ALA mutation were excluded from quantification in channels containing WT and ASP-expressing samples and vice versa). This was done to prevent TMT-induced bleed-through affecting the quantification. Constructs with low expression levels (<20% of the median expression level of all constructs in that set) were also removed, as well as peptides mapping to GFP, since some constructs also produced free GFP as seen on immunoblots. To fit degradation constants ($k_{deg}$) signal sums of the remaining peptides of each construct were calculated and fitted with a linear fit over the linear portion of the time course (the last time point was removed for constructs with rapid degradation).

To estimate statistically, whether there are differences in degradation rates, fold changes of the signal sum of each mutant over the corresponding wild-type (within each set) were calculated for each time point. Linear fits on fold changes against time were done using the `lm()` function in R and resulting probabilities of non-zero slope (Pr_t) (indicating, whether there's a difference in slope between the mutant and the corresponding wild-type) were corrected for multiple testing using the Benjamini–Hochberg method in the function `p.adjust()`.

**Co-pull-down interactome analysis.** Changes in the tight interactome of the mutants compared to wild-type were estimated from the IsobarQuant protein output file at time point zero (before label switch). Briefly, proteins were filtered as described above, and signal sum values were normalized to a GFP-only channel in each set. For PSMA5, differences in interactions were verified in a separate pull-down experiment without SILAC pulse in triplicate, and differences in interaction were identified by applying a limma analysis[45] on the fold changes.

**Reporting summary**
Further information on research design is available in the Nature Portfolio Reporting Summary linked to this article.

## Data availability
The mass spectrometry proteomics data generated in this study have been deposited in the ProteomeXchange Consortium database via the PRIDE[46] partner repository under the following accession codes PXD033254 (PPToP proteome-wide exploratory dataset) and PXD032945 (AP-MS validation dataset of GFP fusion proteins). The processed data including peptide-level statistics used in this study are provided in the Supplementary information. The quantitative models and their behavior in a pSILAC experiment as well as the experimental data presented in this study can be browsed and visualized via an interactive web application at https://apps.embl.de/pptop. Source data for figures showing quantitative data are provided with this paper. Reference data used in this study are accessible as follows:

- Uniprot database: https://www.uniprot.org/

- Annotation of phosphosites from ref. 14: Supplementary Table 3 of original publication
- Phosphoproteomics data along the cell cycle from ref. 22: Supplementary Material of original publication
- Predicted degrons from ref. 25: http://degron.phasep.pro/
- Verified degrons from ref. 26: http://dosztanyi.web.elte.hu/CANCER/DEGRON/TP.html
- Structure of PSMA5 in the proteasome: PDB ID 6MSB
- Database of core protein complexes at CORUM (https://mips.helmholtz-muenchen.de/corum/), release September 2018 [http://mips.helmholtz-muenchen.de/corum/download/releases/old/corum_2018_09_03.zip]
- D2P2 database of intrinsically disordered proteins: https://d2p2.pro/
- Occupancy estimates for phosphosites from ref. 32: Supporting information of original publication
- Number of studies for each phosphosite: PhosphositePlus (https://www.phosphosite.org/homeAction.action)
- Subcellular localizations of proteins: Human Protein Atlas (https://www.proteinatlas.org)
- Subcellular fractionation phosphoproteomic data from ref. 31: Supplementary Data 5 of original publication. Source data are provided with this paper.

## Code availability

Code for analysis of theoretical models as well as code for fitting experimental data to the theoretical models is available at https://doi.org/10.5281/zenodo.7313879.

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

## Acknowledgements

We thank Dr. Sindhuja Sridharan for valuable and critical discussions regarding the manuscript, Dr. Nils Kurzawa for assistance with the comparative F-statistic approach, Dr. Maria Zimmermann-Kogadeeva for fruitful discussions on model analysis, Dr. Gerd Simon Schmidt for helpful discussions on dynamical systems theory, Cecilé Le Sueur for excellent discussions regarding various statistical analyses, the EMBL Proteomics Core Facility, particularly Mandy Rettel, for outstanding technical assistance with the mass spectrometry measurements and instrumentation, as well as all members of the Savitski and Beck teams for feedback and input. This work was supported by the European Molecular Biology Laboratory. H.M.H. and C.M.P. were supported by a fellowship from the EMBL Interdisciplinary Postdoctoral (EI3POD) Programme under Marie Skłodowska-Curie Actions COFUND (grant number 664726).

## Author contributions

Conceptualization: H.M.H. and M.M.S.; visualization, data curation, and investigation: H.M.H.; formal analysis: H.M.H. and E.-M.G.; mathematical analysis of models: E.-M.G.; methodology: H.M.H. and C.M.P.; writing—original draft, H.M.H. and E.-M.G.; writing—review & editing: H.M.H., E.-M.G., M.B., and M.M.S.; project administration: H.M.H.; Resources and Supervision: M.B. and M.M.S.

## Funding

## Competing interests

The authors declare no competing interests.
