## [Peer Review File · Nature Communications]

REVIEWER COMMENTS

Reviewer #1 (Remarks to the Author):

In this manuscript, Hammarén et al. investigate the results of post-translationally modified peptides in metabolic labeling experiments. I find the results novel and somewhat surprising. Nevertheless, the authors have done an excellent job laying out their arguments, and their reasoning is well substantiated. I believe this is important work that will advance our ability to study the dynamics of post-translational modifications but will also provide important background information for the interpretation of “standard” protein turnover studies. In my opinion, no further experiments or analyses are needed before publication. However, the paper lacks a bit in presentation and clarity. I would encourage the authors to spend some time polishing the manuscript. Below are some minor suggestions that might help with this.

Minor suggestions:

“Protein-Peptide Turnover Profiling reveals wiring of phosphorylation during protein maturation”

I find “wiring” in the title confusing. I guess it is supposed to replace “the relative order of PTM addition and removal along a protein’s lifetime” from the abstract because that term is too long. But I don’t think “wiring” does this for people who have not read the paper.

Perhaps use something like: “Protein-Peptide Turnover Profiling reveals the order of PTM addition and removal (during protein maturation)”?

In Fig. 1B on the right, the formula seems odd: “ $\ln(\text{old}/\text{new}+\text{old}) = \text{old remaining}$ ”. “old remaining” should probably be part of an explanation or title rather than part of a formula.

In 1B: can you provide some intuition in the caption why φ for modified proteins can be different from total? (Please ignore this if you change the order below or point towards C,D)

Alternatively, to my previous comment: Move B after C and D => I believe C and D can be intuitively understood and provides the explanation for B. I was struggling with B before seeing C/D.

In C: What does Model 0:1 stand for? Explain in figure (or text) or use a more intuitive name?

Fig. 1E: What is a biological example of a protein synthesized in a modified form? Explain text and potentially figure legend. Alternatively, it might come down to semantics on what the modified and non-modified version is. I suggest removing E or moving it to the supplement. Conceptionally, E does not seem to provide much new.

Model naming 1:0, 1:2, 1:2r seems odd and non-intuitive.

In figure 2E: is the y-axis multiple hypotheses corrected?

Fig2H: what do * and *** mean? Please spell out p-values or similar.

It took me a while to decipher Fig. 3B. Edit Fig 3B legend to something like? "Unmodified peptides with faster clearance are strongly enriched for the same peptide also detected in its phosphorylated form. Fisher's exact test, $p < 2.2e-16$."

Figure 5 legend: "Pubmed and PhosphoSitePlus were accessed on 16.11.21 and searched using the gene name or common alternative names, where applicable." This seems more appropriate for the supplement rather than the figure legend.

Line 98

The name Model 2:1 comes out of nowhere in the text or the figure legend. Model 1:0 is not explained in the text at all. => remove model name, explain meaning of e.g. 2:1 or remove model names.

120 " Arguably, the simplest solution is to relax the assumption that a protein is synthesized in an unmodified form."

" This can biologically be interpreted to represent modification during or immediately after translation."

These two sentences seem to contradict each other.

Reviewer #2 (Remarks to the Author):

The manuscript by Hammeren and co-workers has used theoretical kinetic modeling combined with experimental pulsed stable isotope labeling of amino acids in cell culture (pSILAC). The analysis has focused on the influence of protein phosphorylation on the stability of proteins in HeLa cells. They show that metabolic labeling in combination with PTM-specific enrichment does not measure the effects of PTMs on protein stability. Rather, it indicates the relative order of addition and removal of PTMs over the life of a protein. As an example, they analyzed in more detail the phosphorylation sites on cell cycle-specific proteins (MKI67) and protein complex formation intermediates (PSMA5). The results provide a new view of the function of PTMs by placing them in the context of a protein's stability. Overall, the manuscript is well-written and focuses on theoretical aspects for the calculation of protein turnover. However, it remains a bit unclear how the authors have implemented the cell cycle in the calculation of turnover rates (k_{deg}). A short explanation might be helpful. Do the authors have observed proteins with a phosphorylated PEST domain (phosphodegron) to be faster degraded? The finding that the differences in clearance are independent of the effects of protein stability is interesting. However, the authors only tested tagged and overexpressed proteins. One might consider using here endogenous proteins modified by Crispr/Cas for the corresponding phosphorylation sites. It might be that the interaction and localization of these proteins do not reflect the "real" function and stability of the endogenous protein. To validate the theory for the interaction of the non-phosphorylated PSMA5 protein with other proteasomal subunits, I recommend designing an experiment to dephosphorylate/phosphorylate endogenous (wild-type) PSMA5 and then demonstrate interaction/incorporation into the mature proteasome complex. An in-vitro experiment would be an alternative approach. Since the SRSF2 part is only briefly explained in the results section (the last two sentences and Figure S6), it might be better to take this part out.

Minor

Figure 1 G and 1F are mentioned in the text but not present in the figure. The SIFT score should be briefly explained.

Reviewer #3 (Remarks to the Author):

This is a well-written manuscript with a fascinating and straight forward theoretical basis for understanding a large body of experimental results. The authors test whether a change in peptide turnover observed with post-translational modification can teach us anything about the impact of protein post-translational modifications.

The work represents an important contribution to the scientific literature in that it attempts to provide a framework for future studies looking into the use of protein modification in the protein maturation process.

The message can be strengthened and improved with some changes to the current manuscript. The theoretical basis is sound, but the question is whether the data supports the PTM effect as a general part of the maturation process or not.

Comment on the Fig S2B 5-hour time point max median deviation for modified peptides. Also, comment on how the range for both phospho and unmodified peptide continually increases in Figure S2B. This deviation could be explained biologically or analytically. Does this inflection point at 5 hours disappear if using peptides that are present in all time points?

I recognize that the phosphodeficient and phosphomimetic mutants represent a large amount of work, but the analysis/result is quite underwhelming. As the authors note, if phosphorylation is a key step in the maturation then the ala mutant should significantly disrupt maturation by preventing the first step and the Glu mutant should inhibit the phospho removal dependent step. The explanation of how this is connected to proteolytic stability is not sufficient. Why was a huge GFP fusion used at all instead of a small affinity tag? The PSMA5 mutants do not pull down PSMA1 in the example and do not affect turnover rate. Does this suggest that the processing is not really phospho-dependent? Or are the GFP-fusion proteins all simply excess to normal cellular processes and functions in the cell since the endogenous protein is still present? In the current form, this data does not contribute and confuses the message.

I believe the other data still carries the message, but as the authors state at the end of Figure 5 Legend. 'Comparison of differences in clearance rates from PPToP (Δk_{app}) and wild-type-to-mutant differences from the exogenous expression experiment show no significant correlation (see also Figure S5 B), indicating that differences in clearance measured from PPToP are not predictive of differences in protein degradation caused by the PTM' The easiest explanation is that this is an artifact of the fusion protein over expression. This statement also seems to contradict the entire message of the manuscript.

I recommend moving Figure 5 to the supplement and using this space to expand the very interesting analysis summarized in 6F and 6G. Also consider reformulating Figure 5A to make it rank-ordered on the PPToP T1/2 or another metric connected to the hypothesis of the paper, similar to Fig 5D. Fig 5A in its current form makes it harder to evaluate any trends (or lack thereof).

Minor:

Abstract line 30 'does not' seems to be too strong perhaps stated as 'an equally valid explanation' would be better

line 75: 'allow deriving hypotheses' is difficult to understand. Perhaps you mean 'allow hypotheses on ... to be tested?'

line 82: 'faster clearance as expected' should be 'faster clearance than expected'

line 112: The statement 'the entire protein' is vague. Do you mean the entire protein sequence or the total protein pool?

Figure 2 legend: no description of panel D. also DNAJC2 has both faster and slower clearance. If this is real it is worth a discussion point as to the separation of this total protein pool into faster and slower modified subpopulations

Line 148: The trend in Fig S2A shows a very interesting trend with an apparent maximum median ratio at 5 hours. Does this trend reflect the different populations of phospho sites at each time point or do you see phosphosites which exhibit this equilibrium style transient difference?

REVIEWER COMMENTS

Reviewer #1 (Remarks to the Author):

In this manuscript, Hammarén et al. investigate the results of post-translationally modified peptides in metabolic labeling experiments. I find the results novel and somewhat surprising. Nevertheless, the authors have done an excellent job laying out their arguments, and their reasoning is well substantiated. I believe this is important work that will advance our ability to study the dynamics of post-translational modifications but will also provide important background information for the interpretation of “standard” protein turnover studies. In my opinion, no further experiments or analyses are needed before publication. However, the paper lacks a bit in presentation and clarity. I would encourage the authors to spend some time polishing the manuscript. Below are some minor suggestions that might help with this.

We thank the reviewer for this positive feedback, and agree that the presentation of the manuscript can be improved to clarify the main message. Please see below for our point-by-point responses.

Minor suggestions:

“Protein-Peptide Turnover Profiling reveals wiring of phosphorylation during protein maturation”

I find “wiring” in the title confusing. I guess it is supposed to replace “the relative order of PTM addition and removal along a protein’s lifetime” from the abstract because that term is too long. But I don’t think “wiring” does this for people who have not read the paper. Perhaps use something like: “Protein-Peptide Turnover Profiling reveals the order of PTM addition and removal (during protein maturation)”?

We are grateful of the proposed title change and have adopted it as suggested.

In Fig. 1B on the right, the formula seems odd: “ $\ln(\text{old}/\text{new}+\text{old}) = \text{old remaining}$ ”. “old remaining” should probably be part of an explanation or title rather than part of a formula.

Indeed, the layout was confusing. We have modified this part of Fig. 1B to be more explanatory.

In 1B: can you provide some intuition in the caption why ϕ for modified proteins can be different from total? (Please ignore this if you change the order below or point towards C,D)

Fig. 1B was referring to the fact that a metabolic pulse experiment like the one described in Fig. 1A can generate differing clearance profiles for total and modified species, as has been shown in previous experiments. We have added this information to the figure legend.

Alternatively, to my previous comment: Move B after C and D => I believe C and D can be intuitively understood and provides the explanation for B. I was struggling with B before seeing C/D.

Fig. 1B is intended as an introduction into the possible shapes that clearance curves can take in experimental data. This has been clarified in the legend (see also response to comment above).

In C: What does Model 0:1 stand for? Explain in figure (or text) or use a more intuitive name?

We apologise for forgetting to explain our model nomenclature. We have added the following explanation to the figure legend: "Models are named "Model x:y" (e.g., Model 0:1) to indicate the number of modified species (x), and the number of all species altogether (y)."

Fig. 1E: What is a biological example of a protein synthesized in a modified form? Explain text and potentially figure legend. Alternatively, it might come down to semantics on what the modified and non-modified version is. I suggest removing E or moving it to the supplement. Conceptionally, E does not seem to provide much new.

This is an important point, and we thank the reviewer for bringing this up. As mentioned in the main text describing the reverse Model 1:2r: "This can biologically be interpreted to represent modification during or immediately after translation." However, we did not mention any examples. We have thus now pointed out that this could represent cotranslational modification as well as included four references, showing examples of cotranslational modification with O-GlcNAc, ubiquitin, as well as phosphorylation.

We'd wish to keep Model 1:2r as part of the main figure, as it brings across an important point: a straight-forward explanation for faster clearance is to change the relative order of the modified and unmodified species.

Model naming 1:0, 1:2, 1:2r seems odd and non-intuitive.

We have added an explanation to the figure legend (see also above).

In figure 2E: is the y-axis multiple hypotheses corrected?

Figure 2E shows a volcano-like plot of the statistical comparison of clearance profiles of individual peptides and the respective unmodified protein median. The y axis represents the F-statistic from this comparison, which by nature is not yet corrected for multiple testing. However, for hit calling (and thus also colouring in Fig. 2E), we have converted the F-statistic to a p value, which is then corrected for multiple testing using the Benjamini-Hochberg method.

From the methods part of the manuscript:

"An F-statistic was calculated for fitting the spline to either both peptide and protein median together (H0 model) or to each separately (H1 model). [...] The thusly calibrated F-statistic distribution across both fractions was used to calculate p values for each peptide using the pf() function from the stats package in R and corrected for multiple testing using Benjamini-Hochberg correction. Cases with adjusted p value <= 0.001 were considered as hits."

Fig2H: what do * and *** mean? Please spell out p-values or similar.

We thank the reviewer for pointing out this oversight. Visualisations of all statistical tests across the manuscript have been overhauled with p values explicitly spelled out and other relevant information about the statistical tests added to figure legends in line with the journal's guidelines. We have additionally added more information on the statistical tests used to the Method section, including references to each relevant figure panel with statistical tests.

It took me a while to decipher Fig. 3B. Edit Fig 3B legend to something like? "Unmodified peptides with faster clearance are strongly enriched for the same peptide also detected in its phosphorylated form. Fisher's exact test, $p < 2.2e-16$."

We appreciate this suggestion and have modified the text in the Figure legend accordingly. To further clarify mapping of phosphosites and identifying unmodified and modified peptides carrying the same phosphosites (without unnecessarily cluttering the main text), we have added a paragraph in the methods section entitled:

"Matching phosphorylated and unphosphorylated peptides."

Figure 5 legend: "Pubmed and PhosphoSitePlus were accessed on 16.11.21 and searched using the gene name or common alternative names, where applicable." This seems more appropriate for the supplement rather than the figure legend.

In restructuring Figure 5, this panel has been moved to the supplementary, and the sentence in question has been moved to the expanded Methods section.

Line 98

The name Model 2:1 comes out of nowhere in the text or the figure legend. Model 1:0 is not explained in the text at all. => remove model name, explain meaning of e.g. 2:1 or remove model names.

We apologise for forgetting to explain our model nomenclature. We have added the following explanation to the figure legend: "In the naming of the models (e.g., Model 0:1), "0" stands for the number of modified species, "1" for the number of species altogether."

120 " Arguably, the simplest solution is to relax the assumption that a protein is synthesized in an unmodified form."

" This can biologically be interpreted to represent modification during or immediately after translation."

These two sentences seem to contradict each other.

We agree that the wording here should have been more precise and direct. We have altered the main text referring to this (see also comment to Fig. 1E above) as follows:

"Arguably, the simplest solution is to reverse the relative order of the two species (Model 1:2r, Figure 1g). This can biologically be interpreted to represent modification during or immediately after translation (i.e. cotranslationally) (Wang et al. 2013; Duttler et al. 2013; Zhu et al. 2015; Oh et al. 2010)."

Reviewer #2 (Remarks to the Author):

The manuscript by Hammeren and co-workers has used theoretical kinetic modeling combined with experimental pulsed stable isotope labeling of amino acids in cell culture (pSILAC). The analysis has focused on the influence of protein phosphorylation on the stability of proteins in HeLa cells. They show that metabolic labeling in combination with PTM-specific enrichment does not measure the effects of PTMs on protein stability. Rather, it indicates the relative order of addition and removal of PTMs over the life of a protein. As an example, they analyzed in more detail the phosphorylation sites on cell cycle-specific proteins (MKI67) and protein complex formation intermediates (PSMA5). The results provide a new view of the function of PTMs by placing them in the context of a protein's stability. Overall, the manuscript is well-written and focuses on theoretical aspects for the calculation of protein turnover.

However, it remains a bit unclear how the authors have implemented the cell cycle in the calculation of turnover rates (k_{deg}). A short explanation might be helpful.

We thank the reviewer for pointing out that the cell cycle correction done should be referred to more clearly in the text. We have thus added the following comment to the Results section:

“To estimate clearance of peptides, we first corrected the raw SILAC data for dilution due to cell division (see Methods for details) giving estimates of fraction of old protein remaining (φ).”

This refers to the Materials and methods, which states:

“Cell cycle times for each replicate were estimated from the protein median values in the unmodified fraction. Assuming exponential decay of most proteins, we have the linear relationship¹:

$$\ln\left(\frac{\text{new}}{\text{old}} + 1\right) = (k_{deg} + \frac{\ln 2}{t_{cc}})t$$

where $\frac{\text{new}}{\text{old}}$ is the SILAC ratio of new and old protein, k_{deg} is the protein-specific degradation constant, and t_{cc} the cell cycle time. We estimated t_{cc} from the 1% longest-lived proteins where, assuming no active protein degradation ($k_{deg} = 0$), the slope is defined by $\frac{\ln 2}{t_{cc}}$, from which we got t_{cc} estimates of 28.0 h, 26.5 h, 27.0 h, and 22.2 h for replicates 1 through 4, respectively. Subsequently, new/old SILAC ratios were transformed into fraction of old remaining (see Theory Supplement for reasons of doing so), and corrected for cell cycle as follows:

$$\text{corrected old remaining} = \varphi = \ln\left(\frac{\text{new}}{\text{old}} + 1\right)^{-1} + \frac{\ln 2}{t_{cc}}t$$

“

The Theory Supplement further details the theoretical reasons for using a cell division-corrected clearance rate in section: “3.4 A single homogeneous protein pool - Correction for cell cycle related increase in protein abundance”

Do the authors have observed proteins with a phosphorylated PEST domain (phosphodegron) to be faster degraded?

This is a valuable suggestion to cross-validate our theory-based prediction that hits with differing clearance in a SILAC PPToP experiment should not correspond to proteoforms with differing proteolytic stability, but rather the order of modification events.

To this end, we have included a comparison to predicted and verified degron sequences. We opted to not only focus on PEST sequences specifically, as more recent analyses (Meszaros et al., 2017, PMID: 28292960) of degrons have significantly expanded from the initial PEST hypothesis (Rogers et al, 1986). We thus used a machine learning-based dataset of predicted degrons (Hou et al., 2022, PMID: 35836176), as well as a curated set of verified degron sequences (Meszaros et al., 2017).

The results have been added as Figure 5 c and d. These show no significant enrichment of sequences proximal to predicted degrons in our PPToP phosphohits, nor any significant overlap with known degrons.

This analysis thus corroborates our theoretical prediction that hits with differing clearance should not be interpreted as representing degrons (PTM-controlled or not).

The text has been updated to incorporate this analysis.

The finding that the differences in clearance are independent of the effects of protein stability is interesting. However, the authors only tested tagged and overexpressed proteins. One might consider using here endogenous proteins modified by Crispr/Cas for the corresponding phosphorylation sites.

Although intuitively, our finding that apparent differences in clearance are unrelated to proteolytic stability might be surprising (as it was to us initially), it is fully in line with our theoretical considerations and mathematical modelling.

The reason for opting for a tag-based approach was to allow testing a sufficiently large number of phosphorylation sites to test our hypothesis as comprehensively as necessary. While admittedly, CRISPR/Cas-based genetic engineering of the respective endogenous loci would have represented a cleaner experiment, the generation 70+ individual genetically-engineered cell lines was clearly out of scope given the resources available to us.

Furthermore, our follow-up analysis is based on measuring the relative stability difference between each mutant and the respective wild-type construct. As both the mutant and the wild-type carry the same tag and are both exogenously expressed, any difference measured should stem from the mutation specifically. Expression levels are also a point to consider, which is why we have verified that our different constructs are expressed at comparable levels in the cell (Figure S6 a).

It might be that the interaction and localization of these proteins do not reflect the “real” function and stability of the endogenous protein.

This is an important point, as adding fusion tags to proteins can alter their in-cell behaviour. To address this, we have leveraged the fact that we used a monomeric EGFP-tag and done microscopy to verify that our exogenously expressed constructs localise correctly. This data

has been added as a new Figure S6 b showing correct localisation for 22 of the 23 target proteins tested. The single construct that exhibited incorrect localisation was excluded from further analysis. The text in the Results section has been amended accordingly.

We thus thank the reviewer for this suggestion, as it allowed us to find and remove a potentially misleading datapoint from our analysis by spotting a protein, where addition of the EGFP-tag caused non-physiological behaviour.

To validate the theory for the interaction of the non-phosphorylated PSMA5 protein with other proteasomal subunits, I recommend designing an experiment to dephosphorylate/phosphorylate endogenous (wild-type) PSMA5 and then demonstrate interaction/incorporation into the mature proteasome complex. An in-vitro experiment would be an alternative approach.

We agree that following up our finding of S16 as a potentially interesting site for PSMA5 and its incorporation into the proteasome would benefit from more corroborating evidence.

However, a fully reductionistic in vitro experiment would require in vitro reconstitution of the entire proteasome consisting of 14 individual subunits as well as 5 dedicated chaperone proteins (an undertaking which was only recently successfully demonstrated for the first time (Rego et Fonseca, 2019, PMID: 31473102)), and is outside of the scope of this study.

We have thus opted for interrogating multiple lines of other evidence to further investigate the role of PSMA5 phosphorylation in its incorporation into the proteasome. Firstly, we expanded our initial PSMA5 pull-down experiments to also include a construct with only the single point S16A mutation (in addition to the S16, S56 double mutants), which likewise showed reduced interaction with PSMA1 compared to wild-type PSMA5 (new Figure S8 a). Secondly, we performed pull-down analyses under less stringent conditions than the RIPA buffer previously used. Pull-downs in detergent-free buffer with mild lysis conditions (freeze-thaw) as well as pull-downs in the presence of mild detergents (NP40) both showed not only a lessened interaction of mutated PSMA5-GFP with PSMA1 compared to wild-type PSMA5-GFP, but further also showed a significant reduction in pull-down efficiency of other proteasomal proteins (Figure S8 a) corroborating the role of S16 phosphorylation for the correct assembly of PSMA5 into the proteasome.

Thirdly, we investigated phosphoproteomic data from HeLa subcellular fractionation experiments (Martinez-Val et al, Nat Comm 2021, PMID: 34876567) to assess how endogenous phosphorylated PSMA5 partitions into the different subcellular fractions (Figure S8 b). We found that while proteasomal proteins (including PSMA5) can be detected in all subcellular fractions (including less soluble fractions representing larger assemblies as well as membrane-bound etc proteins), PSMA5 phosphorylated on S16 is only measured in the most soluble fraction, in line with the phosphorylated form being present only in free PSMA5.

Since the SRSF2 part is only briefly explained in the results section (the last two sentences and Figure S6), it might be better to take this part out.

While we intended to use the SRSF2 example to point out that PTM sites identified using PPToP can have varying biological consequences (such as changes in protein-protein

interactions), we appreciate that a further, only briefly discussed example is probably more confusing than useful for the clarity of the message. We have thus removed the SRSF2 figure as well as the accompanying text from the results section.

Minor

Figure 1 G and 1F are mentioned in the text but not present in the figure. The SIFT score should be briefly explained.

Figure references have been overhauled and checked.

We have also added the explanation of the SIFTS acronym, which in fact was accidentally misspelled as "SIFT" in our initial manuscript. We are grateful for pointing out these details.

Reviewer #3 (Remarks to the Author):

This is a well-written manuscript with a fascinating and straight forward theoretical basis for understanding a large body of experimental results. The authors test whether a change in peptide turnover observed with post-translational modification can teach us anything about the impact of protein post-translational modifications.

The work represents an important contribution to the scientific literature in that it attempts to provide a framework for future studies looking into the use of protein modification in the protein maturation process.

The message can be strengthened and improved with some changes to the current manuscript. The theoretical basis is sound, but the question is whether the data supports the PTM effect as a general part of the maturation process or not.

We thank the reviewer for this positive feedback, and agree that the presentation of the manuscript can be improved to clarify the main message. We have updated, e.g., the manuscripts abstract to be more explicit and precise in order to prevent potential confusion. We have, for instance, removed the mention of a protein's "lifetime" in the abstract, as this is easily interpreted as referring to degradation times, which is not our intent in this context.

Please see below for our point-by-point responses.

Comment on the Fig S2B 5-hour time point max median deviation for modified peptides.

Also, comment on how the range for both phospho and unmodified peptide continually increases in Figure S2B. This deviation could be explained biologically or analytically. Does this inflection point at 5 hours disappear if using peptides that are present in all time points?

We are grateful for the reviewer for pointing out this detail, as it revealed an unfortunate artefact of the median normalisation. Indeed, the apparent maximum was due to different subpopulations of (phospho)peptides being present in the different time points. We have thus modified Figure S2 b to show unnormalised values, which should represent the data more faithfully. We have also opted to show the distribution as density ridgeplots, as they more clearly show the broader distribution of phosphopeptide clearance.

Furthermore, and more importantly, our careful reanalysis of Figure S2 also convinced us to reassess the data from the 28 h time point and its place in the time series. The 28 h time point is far removed from the rest of the series (which are otherwise equidistant from each other), and we have thus decided to remove it from our downstream analysis. While this has slightly lowered the overall number of peptides/proteins we can statistically analyse (as our method requires ≥ 4 time points), we believe it does increase the robustness of our analysis. It also does not impact our conclusions.

The text has been changed to reflect this.

I recognize that the phosphodeficient and phosphomimetic mutants represent a large amount of work, but the analysis/result is quite underwhelming. As the authors note, if phosphorylation is a key step in the maturation then the ala mutant should significantly

disrupt maturation by preventing the first step and the Glu mutant should inhibit the phospho removal dependent step. The explanation of how this is connected to proteolytic stability is not sufficient.

We thank the reviewer for acknowledging the major effort that the validation work with a library of mutant constructs represents! However, we respectfully disagree with the reviewer regarding the expected result from this experiment. As we have shown theoretically (outlined in Figure 1, and discussed in depth in the Theory Supplement), a difference in clearance (as measured by SILAC labelling) of a PTM-modified protein species should not be due to any effect on the protein's proteolytic stability. As a consequence, mutations of the thus identified sites are not expected to change the protein's degradation rate. This is exactly what our experiment shows.

Thus, while we are aware that it may not be initially intuitive that measured differences in clearance are unrelated to proteolytic stability, it is fully in line with our theoretical consideration, mathematical modelling and validation experiments. The observed phenomenon stems from other effects than protein stability, namely mostly temporal distance from the protein's synthesis.

Why was a huge GFP fusion used at all instead of a small affinity tag?

This is an important point, as large fusion tags to proteins can alter their in-cell behaviour. We primarily opted for a GFP tag to enable pull-downs using the efficient GFP-Trap system, as well as to future-proof our sizable construct library for potential future experiments.

Furthermore, we have now leveraged the GFP-tag and done microscopy to verify that our exogenously expressed constructs localise correctly. This data has been added as a new Figure S6 b showing correct localisation for 22 of the 23 target proteins tested. The single construct that exhibited incorrect localisation was excluded from further analysis. The text in the Results section has been amended accordingly.

We thus thank the reviewers for this and similar suggestions, as it allowed us to find and remove a potentially misleading datapoint from our analysis by spotting a protein, where addition of the EGFP-tag caused non-physiological behaviour.

The PSMA5 mutants do not pull down PSMA1 in the example and do not affect turnover rate. Does this suggest that the processing is not really phospho-dependent? Or are the GFP-fusion proteins all simply excess to normal cellular processes and functions in the cell since the endogenous protein is still present? In the current form, this data does not contribute and confuses the message.

These are also important points, and we'll discuss them separately. Our analysis of the effect of each mutation is always in relation to a similarly-tagged wild-type (i.e. unmutated) construct. This should remove potential artefacts caused by the tagging procedure, as only the mutated site differs between the constructs. Expression levels are also a point to consider, which is why we have verified that our different constructs are expressed at comparable levels in the cell (Figure S6 a).

In the case of PSMA5, the wild-type construct does still interact with PSMA1, and we thus deem it unlikely that the effects we see between the wild-type and the mutant constructs would be due to the use of the EGFP tag, or unphysiological expression levels.

However, we do agree that following up our finding of S16 as a potentially interesting site for PSMA5 and its incorporation into the proteasome would benefit from more corroborating evidence. We have thus interrogated multiple lines of other evidence to further investigate the role of PSMA5 phosphorylation in its incorporation into the proteasome. Firstly, we expanded our initial PSMA5 pull-down experiments to also include a construct with only the single point S16A mutation (in addition to the S16, S56 double mutants), which likewise showed reduced interaction with PSMA1 compared to wild-type PSMA5 (new Figure S8 a).

Secondly, we performed pull-down analyses under less stringent conditions than the RIPA buffer previously used. Pull-downs in detergent-free buffer with mild lysis conditions (freeze-thaw) as well as pull-downs in the presence of mild detergents (NP40) both showed not only a lessened interaction of mutated PSMA5-GFP with PSMA1 compared to wild-type PSMA5-GFP, but further also showed a significant reduction in pull-down efficiency of other proteasomal proteins (Figure S8 a) corroborating the role of S16 phosphorylation for the correct assembly of PSMA5 into the proteasome.

Thirdly, we investigated phosphoproteomic data from HeLa subcellular fractionation experiments (Martinez-Val et al, Nat Comm 2021, PMID: 34876567) to assess how endogenous phosphorylated PSMA5 partitions into the different subcellular fractions (Figure S8 b). We found that while proteasomal proteins (including PSMA5) can be detected in all subcellular fractions (including less soluble fractions representing larger assemblies as well as membrane-bound etc proteins), PSMA5 phosphorylated on S16 is only measured in the most soluble fraction, in line with the phosphorylated form being present only in free PSMA5.

Finally, as discussed above, our theoretical analysis shows that we should not be expecting a difference in degradation (at least directly), when mutating these sites with differing dynamics. While it is conceivable that constructs that can not get incorporated into growing complexes, for instance, could be targeted by the cellular degradation machinery, this is not what one should primarily expect, and for the case of PSMA5, also not what our data suggests.

I believe the other data still carries the message, but as the authors state at the end of Figure 5 Legend. 'Comparison of differences in clearance rates from PPToP (Δk_{app}) and wild-type-to-mutant differences from the exogenous expression experiment show no significant correlation (see also Figure S5 B), indicating that differences in clearance measured from PPToP are not predictive of differences in protein degradation caused by the PTM' The easiest explanation is that this is an artifact of the fusion protein over expression. This statement also seems to contradict the entire message of the manuscript.

The major message that we want to bring across is that the apparent turnover rates reveal the order of PTM addition and removal.

As mentioned above, we realise how unexpected the theoretical prediction of apparent clearance rates and cellular degradation rates can intuitively seem. Yet, our experimental

results also support this conclusion. Nevertheless, we have now updated the manuscript in order to clarify and crystallise this message more clearly, where our initial manuscript might have been unclear.

See also discussion above regarding the use of exogenous expression of GFP-fusion constructs as well as considerations on the physiological function and localisation of the constructs used.

I recommend moving Figure 5 to the supplement and using this space to expand the very interesting analysis summarized in 6F and 6G. Also consider reformulating Figure 5A to make it rank-ordered on the PPToP T1/2 or another metric connected to the hypothesis of the paper, similar to Fig 5D. Fig 5A in its current form makes it harder to evaluate any trends (or lack thereof).

We have reordered Figure 5 as suggested and moved it to the supplement, as it mainly serves to introduce our experimental mutant constructs and provide additional information.

To expand our discussion, as well as to highlight more clearly what a PPToP dataset with high time-resolution such as ours can provide, our updated manuscript now includes direct fitting of a simplified theoretical model to the data. This shows that we can measure biologically meaningful differences in writing and erasing rate constants, which we demonstrate by showing distinct behaviour of N-terminal acetylation and phosphorylation sites. Furthermore, elucidation of the rate constants also allows estimation of the stoichiometry of a modification (a.k.a. occupancy). Comparison to previous data shows that our dynamics-derived occupancies are in line with experimental estimates (Lim et al., JPR 2017, PMID: 28985074).

These analyses have been added as a new section to Results and an accompanying Figure 7.

Minor:

Abstract line 30 'does not' seems to be too strong perhaps stated as 'an equally valid explanation' would be better

We appreciate the suggestion to soften our language in the abstract. However, we feel that this phrasing captures the essence of our manuscript. Already our theoretical analysis shows very clearly that the observed difference in clearance of PTM-modified peptide species should not be interpreted as a difference in proteolytic stability. Notably, this does not invalidate the use of metabolic labelling to measure protein turnover/degradation, when applied to an entire pool of protein (i.e. all proteoforms). We have added a notion to the discussion to indicate this.

"It should also be noted that these conclusions do not in any way invalidate the use of pSILAC to measure protein degradation, when applied to an entire pool of proteoforms of a protein (Schwanhäusser et al. 2011; Mathieson et al. 2018). Rather, our conclusions

highlight the need to more careful consideration, when measuring introduction of a metabolic label into a network of interconvertible species, such as proteins undergoing modifications by PTMs.”

We hope that the reviewer agrees with us, that our reference to previous work is sufficiently careful.

line 75: 'allow deriving hypotheses' is difficult to understand. Perhaps you mean 'allow hypotheses on ... to be tested?'

We have changed the text to be simpler using “suggests”.

line 82: 'faster clearance as expected' should be 'faster clearance than expected'
We have changed the text to remove this potentially confusing part.

line 112: The statement 'the entire protein' is vague. Do you mean the entire protein sequence or the total protein pool?

We agree that the choice of nomenclature regarding the words “total protein”, “entire protein”, and “entire protein pool” were unclear and at times inconsistently used.

We have now overhauled the manuscript to use distinct terminology for these terms as follows:

“Total” - flow-through fraction of the phosphoenrichment procedure. Consists of mostly unmodified proteins.

“Entire protein pool P” - the mean behaviour of all proteoforms belonging to a single protein group. E.g. in Model 1:2, this refers to the sum of P_p and P_u . In practice, this is measured as the median behaviour of all unmodified peptides mapped to a protein.

Figure 2 legend: no description of panel D. also DNAJC2 has both faster and slower clearance. If this is real it is worth a discussion point as to the separation of this total protein pool into faster and slower modified subpopulations

We are thankful for pointing out these overlooked details. Figure 2 legend has now been corrected.

We believe the example of DNAJC2 to be a real phenomenon, and highlights the richness of the data. Similar situations exist for numerous proteins, which can be accessed and browsed in the shiny web application accompanying the manuscript (<https://apps.embl.de/pptop>).

In the revised manuscript, we removed the DNAJC2 example, as after omission of the 28 h time point (see discussion above), it had too few data points to act as a representative example.

Line 148: The trend in Fig S2A shows a very interesting trend with an apparent maximum median ratio at 5 hours. Does this trend reflect the different populations of phospho sites at each time point or do you see phosphosites which exhibit this equilibrium style transient difference?

The reproducibility distribution at 5 h is indeed slightly broader than at 6 h, but this is simply due to the fact that time points 3 h and 6 h were measured in four biological replicates, whereas the other time points were measured as biological duplicates. This is also reflected in the number of individual peptides in each time point, as we only included peptides measured in a minimum of two replicates.

REVIEWERS' COMMENTS

Reviewer #2 (Remarks to the Author):

The authors have addressed all of the reviewer's questions and the reviewer has no further concerns about the current version of the manuscript. The point by point explanation in the rebuttal letter was clearly stated and has clarified any unclear points.

Reviewer #3 (Remarks to the Author):

The authors have addressed all of my comments in a satisfactory way. I recommend publication.